# GUST: Combinatorial Generalization by Unsupervised Grouping with Neuronal Coherence

**Hao Zheng**[†]    **Hui Lin**[†]    **Rong Zhao**[*]
Center for Brain-Inspired Computing Research,
Optical Memory National Engineering Research Center,
Tsinghua University - China Electronics Technology HIK Group
Co. Joint Research Center for Brain-Inspired Computing,
IDG / McGovern Institute for Brain Research at Tsinghua University,
Department of Precision Instrument,
Tsinghua University, Beijing 100084, China.
`{zheng-h17, linhui21}@mails.tsinghua.edu.cn`
`r_zhao@mail.tsinghua.edu.cn`

## Abstract

Dynamically grouping sensory information into structured entities is essential for understanding the world of combinatorial nature. However, the grouping ability and therefore combinatorial generalization are still challenging artificial networks. Inspired by the evidence that successful grouping is indicated by neuronal coherence in the human brain, we introduce GUST (Grouping Unsupervisely by Spike Timing network), an iterative network architecture with biological constraints to bias the network towards a dynamical state of neuronal coherence that softly reflects the grouping information in the temporal structure of its spiking activity. We evaluate and analyze the model on synthetic datasets. Interestingly, the segregation ability is directly learned from superimposed stimuli with a succinct unsupervised objective. Two learning stages are present, from coarsely perceiving global features to additionally capturing local features. Further, the learned building blocks are systematically composed to represent novel scenes in a bio-plausible manner.

## 1 Introduction

Humans are able to effortlessly group the world in terms of abstract and reusable 'building blocks', and leverage those learned blocks to understand novel contexts. For example, there is a list of evidences that infants as young as 3-month old are able to group information from individual visual elements into wholistic perceptual units [1, 2, 3, 4]. This phenomenon is referred as perceptual grouping [5] or binding in neuroscience and psychology literature [6] and such ability is believed to be fundamental for the compositional generalization of neural network models [7].

However, grouping ability challenges artificial neural networks (ANNs) for several reasons. Firstly, grouping from raw sensory input is in principle an unsupervised task with multiple possible solutions. For example, how to segment a scene[8] or how to chunk a sensory stream into episodic memories[9] do not have a golden criterion. This excludes the solution through massive labeling and supervised learning[10]. Secondly, the representation of symbol-like content (the pattern of a single object) should be able to retain itself and be reused in front of various situations with a common format. Without proper constraints, simple network models can fall into ambiguities when trying to compose

---

[†]Equal contribution.
[*]Corresponding author.
The code is publicly available at: `https://github.com/monstersecond/gust`.

37th Conference on Neural Information Processing Systems (NeurIPS 2023).

those learned patterns, referred to as the binding problem[11]. Thirdly, the cost of representing all possible combinations statically and locally is exponentially high, limiting the capability of the local code to deal with increasingly complex scenes[12, 13]. Lastly, the grouping ability should not be constrained to an explicit prior of separation [14], but can organize itself flexibly and dynamically according to the statistical structure of the input.

In this paper, we take insights from the neural system of the human brain to provide a brain-inspired grouping mechanism that mitigates the aforementioned challenges in ANNs. Neurons in the brain have a spiking nature. The firing activity in the high-dimensional neuronal space can have both spatial and temporal structures. The spatial structure of spike firing, usually referred to as receptive field, is related to certain objects[15] or certain feature content of objects (orientation, color, texture, depth etc.)[16]. Occasionally, the neuronal firings evolve into a coherent state, where large amounts of neurons interrelate in the millisecond timescale (Fig.1b,right)[17]. Specifically, neurons encoding features of the same entity tend to fire synchronously and out of phase with neurons that encode features of different entities[18]. At this time, the temporal structure of the spike firing emerges as the neuronal oscillation arises[19, 20]. In the cognition literature, this coherent state is highly related to awareness and successful percepts [21, 22, 23, 24]. According to these evidences, humans appear to use the spatial and temporal structure of spikes for representing contents and the grouping information among contents, respectively. Salient coherent states are indicators of the successful grouping, which enable better readout of contents by downstream neurons with non-linear dendrites sensitive to coincident events[25]. These insights have been developed as temporal binding theory[26, 27] or correlation brain theory[6] in the neuroscience literature.

Compared with grouping mechanisms in ANNs, grouping with temporal neuronal coherence has several promising advantages. First, the grouping is a flexible and dynamical self-organized process without supervision or explicit prior of the separation. Second, the coherence level may help to encode the grouping uncertainty. Third, grouping in time dimension will naturally maintain a common format[7]. Lastly, the (spatial) representational cost is reduced to a single pool of perceptual elements[7]. However, despite the advantages of grouping with temporal coherence, as far as we know, such grouping mechanism is seldom present in ANNs field due to the gap between temporal binding models and gradient-based ANNs. To be more specific, precise temporal codes rely on spiking neurons, which are non-differentiable and also require simulating the dynamics of neurons. Besides, the connection between neurons is required to be built-up by hebbian-like plasticity rule[28] instead of gradient-based learning rules[29], which necessitates a completely different training framework from that of ANNs. Lastly, it is not clear how neuronal coherence emerge within ANNs framework.

In this paper, we introduce the GUST network (short for **G**rouping **U**nsupervisely by **S**pike **T**iming network) that assigns the grouping information implicitly in spiking synchrony as an emergent property of the network. The coherent state is achieved by iterative bottom-up / top-down processing between a spike coding space (SCS) biased by biological constraints and a denoising autoencoder (DAE)[30, 31] to provide delayed feedback. The top-down delayed feedback architecture for the temporal grouping is closely related to the constructive nature of the brain perception [32]. Notably, the GUST is trained end-to-end by a gradient-based method and therefore compatible with ANNs framework. Besides, the grouping ability is learned implicitly with a general unsupervised objective to directly denoise multi-object visual inputs. Lastly, once the GUST "sees" a given number of objects during training, the acquired grouping ability can be successfully generalized to visual inputs of a different number of objects, showing its combinatorial generalization capability to understand scenes of different structures (eg. number of objects).

Designing GUST is non-trivial for four challenges. First, stochastic dynamics based on spike firing is non-differentiable and gradients cannot be naturally back-propagated as standard artificial networks. Additional biological constraints like refractoriness, non-linear dendrites, transmission delay and timescale difference aggravate the problem. Second, temporal grouping requires relatively high temporal resolution, which in turn implies a long simulation period. Such long simulations challenge the efficiency of backpropagation. Third, as far as we know, there are no trivial unsupervised objective functions to train a temporal grouping network directly from superimposed inputs. Lastly, the representation needs to be explainable in terms of its precise temporal structure. To overcome challenges for back-propagation, we approximate gradients in non-differentiable parts and short-cut the back-propagation by the transmission delay (Section3.5). For unsupervised learning, surprisingly, a super succinct objective function to denoise superimposed visual inputs is shown to enable the GUST to learn to segregate (Section3.4), once biased by a list of biological constraints (eq.1∼eq.7).

For evaluation, we develop methods to quantitatively evaluate neuronal coherence and its relation to the grouping (Section3.6) based on a non-euclidean metric of the precise temporal coding in the visual cortex[33]. By quantitative evaluation and qualitative visualization, the representation in GUST is clearly explained.

In sum, the contribution of GUST is threefold. First, a gradient-based, unsupervised training framework is developed to enable the GUST to group objects with neuronal coherence. Since the framework is compatible with ANNs, it contributes to bridging the temporal binding theory and the ANNs framework. Second, the GUST is the first temporal grouping model that learns to group directly from multi-object inputs and is able to systematically generalize the mechanism to inputs of different number of objects. Third, we develop a clustering method to evaluate the neuronal coherence through a non-linear metric that measures the distance between arbitrary spike trains directly based on their precise temporal structures.

## 2    Grouping in the temporal dimension

**Grouping.** In this work, grouping is formalized as a pixel-level segmentation of visual scenes composed of multiple objects (Fig.1a). It is notable that while only visual inputs are used for clarity, the general mechanism is not restricted to a certain modality. Similarly, if we take the pixel-level elements as more abstract features that are distributed in a high-dimensional representation space, the grouping mechanism is not restricted to the superficial features as well.

**Grouping with neuronal coherence.** Two separated timescales are considered: $\tau_1 \ll \tau_2$ (Fig.1b). During a period of $\tau_2$, we assume that the accumulated firing rate of each neuron reflects certain features of an object. In other words, the spatial structure of a population of neurons encode which feature are present in the current scene. In this work, features are formalized as object-related pixels in an image. When the neuronal coherence state is arrived, the firing of neurons is synchronized within a much narrower temporal window $\tau_1$ so that the tempo-

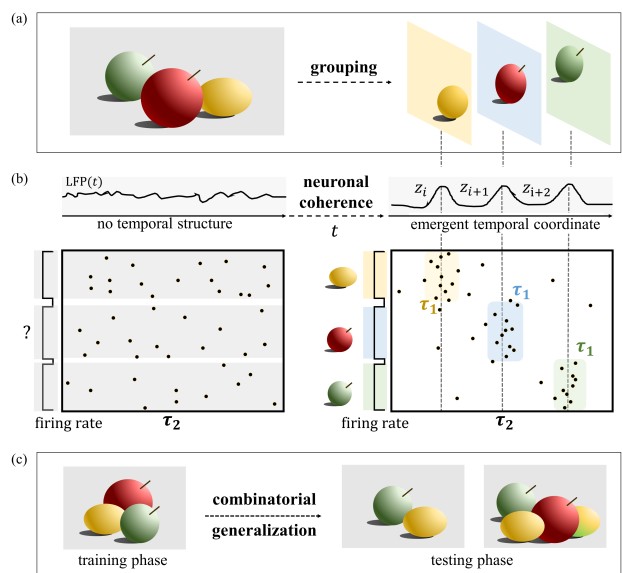

Figure 1:   (a) Grouping as segmentation. (b) Grouping with neuronal coherence. The spike plot shows the emergence of the neuronal coherence (x-axis: time; y-axis: neuronal index). (c) The task scheme for combinatorial generalization

ral structure of spike firing emerges (Fig.1b,right). The temporal structure is reflected in the regular rhythmic pattern of the population activity (LFP), which act as an emergent coordinate ($z_i, z_{i+1}, z_{i+2}$ in Fig.1b ) to assign the additional grouping information. It is notable that the temporal coordinate is not an explicit prior from human experts but an emergent property from the evolution of the network dynamics, Therefore, the grouping is flexible and dynamic.

**Combinatorial generalization by grouping.** If we formalize "seeing" a scene as successfully grouping the visual objects with neuronal coherence, the combinatorial generalization of the grouping is in turn defined as the following question: Given that the GUST has only "seen" scenes composed of a certain number of objects during training, does it generalize its grouping ability to "see" scenes composed of a different number of objects, which structurally change the input distribution(Fig.1c)? In other words, can GUST form coherent neuronal groups even in unseen situations? This question requires the GUST not only learns to group certain types of scenes into building blocks, but also manages to compose the building blocks to understand novel scenes.

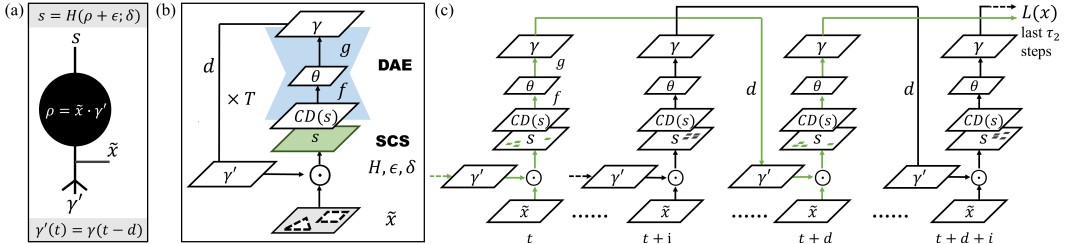

Figure 2: (a) The spiking neuron model. (b) The general architecture of GUST. (c) The unrolling of GUST through time.

## 3 Method

In this section, we first introduce the general architecture of GUST (Section 3.1) and provide details of each module (Section 3.2,3.3). Then, we introduce how three essential challenges are solved including designing the unsupervised objective function (Section 3.4), back-propagation strategy to train the GUST (Section 3.5) and how grouping with neuronal coherence is evaluated (Section 3.6).

### 3.1 General architecture of GUST

How can the coherent states emerge from random firing? While it is still an intricate question without a unified answer, we take inspiration from the anticipating nature of the neural activity in a processing hierarchy. The top-down feedback to the primary sensory cortices carries predictions about the features of the environmental stimuli and such modulatory top-down effects might influence the temporal structure or coherence level of the neuronal responses[32, 34, 35]. Therefore, the predictive feedback acts as positive cues to guide the grouping and coherence and is realized as the reconstructive output of an autoencoder in GUST. Besides, spiking dynamics, as pointed out in neuronal sampling theory[36], provides biological constraints that facilitate bypassing the energy barrier to switch between alternative stable states. In this way, the contents of different groups can get separated in time (See Appendix A.3.1).

The general architecture of GUST is composed of a spike coding space (SCS, green square in Fig.2b) and a denoising autoencoder (DAE, blue shadow in Fig.2b). The grouping solution is represented as neuronal coherence in the SCS and is found by iterative bottom-up neuronal sampling in the SCS and top-down attentional feedback $\gamma$ from the DAE during the whole simulation period $T$. The feedback is delayed by $d$ steps.

### 3.2 Spike coding space (SCS)

The SCS is a layer of the same dimension as input $x$ and is composed of two-compartment pyramidal neuron models (Fig.2a). Each pyramidal neuron receives the driving signals from the noisy visual inputs $\tilde{x}$ (during training[*]) and the delayed modulatory feedback from the DAE output.

$$\gamma_i'(t) = \gamma_i(t - d). \tag{1}$$

where $\gamma_i'$ is the delayed modulatory feedback to neuron $i$, $\gamma_i$ is the output of DAE and $d$ is the delay period. Two sources of inputs are multiplied to determine the noiseless membrane potential $\rho_i(t)$:

$$\rho_i(t) = \gamma_i'(t) \cdot \tilde{x}_i. \tag{2}$$

The stochastic neuron fires a spike if it is not refractory and its membrane potential exceeds a given threshold. More formally,

$$s_i(t) = H(\rho_i(t) + \epsilon; \delta_i(t)). \tag{3}$$

where $\epsilon$ is the random noise drawn from a uniform distribution $U(0, 1)$ and $H$ is the non-linear activation function of the spiking neurons (threshold = 1):

$$H(u_i; \delta_i) = \begin{cases} 1, & \delta_i = 0 \wedge u_i \geq 1. \\ 0, & \delta_i = 0 \wedge u_i < 1. \\ 0, & \delta_i > 0. \end{cases} \tag{4}$$

---

[*]During testing and visualization, it receives original inputs $x$, see Appendix A.7 (Salt&Pepper noise).

where $\delta_i(t)$ is the refractory variable for each neuron, which is non-zero if the neuron is refractory. A neuron falls into refractory period of length $\delta$ if it fires a spike. More formally,

$$\delta_i(t) = \begin{cases} 0, & s_i(t) = 0 \wedge \delta_i(t-1) = 0. \\ \delta_i(t-1) - 1, & s_i(t) = 0 \wedge \delta_i(t-1) > 0. \\ \delta, & s_i(t) = 1. \end{cases} \quad (5)$$

To sum up, the SCS integrates the driving input and the delayed modulatory feedback multiplicatively to determine the spike firing of neurons that are not in the refractory period. Once a neuron fires a spike, it gets into the refractory period that is unable to give another spike firing. The refractory dynamics provides the essential temporal competition to separate groups of different objects and acts as a structural bias for a grouping solution[37].

### 3.3 Denoising autoencoder (DAE)

The coherent activities in SCS are readout by coincident detectors (CD) that integrate adjacent activities within a very narrow time window ($\tau_w = 2$) to feed to subsequent bottom-up processing.

$$s'_i(t) = CD(s_i(t), \cdots, s_i(t - \tau_w + 1))$$
$$= \begin{cases} 1, & \sum_{t'=0}^{\tau_w-1} s_i(t - t') \geq 1. \\ 0, & \sum_{t'=0}^{\tau_w-1} s_i(t - t') < 1. \end{cases} \quad (6)$$

The bottom-up and top-down communication in a processing hierarchical[32] is formulated as a denoising autoencoder that is composed of an encoder $f$, a latent layer $\theta$ and a decoder $g$ (Fig.2b). The encoder extracts more abstract features into the latent representation $\theta$ and the decoder computes the top-down feedback $\gamma$ to SCS. More formally,

$$\gamma_i(t) = g(f(s'_i(t))). \quad (7)$$

The dimension of input and output of DAE are kept the same as SCS and the dimension of the latent space is much smaller to form an information bottle neck[38] of the hierarchical processing.

### 3.4 Loss function

The first challenge is how grouping with neuronal coherence can be learned in an unsupervised way similar to young infants. For this motivation, the objective function should not leak any explicit grouping information.

We train the GUST to actively predict and denoise the multi-object input patterns at timescale $\tau_2$. Only a simple unsupervised loss function is identified:

$$L(x) = (x - \frac{1}{Z} \sum_{t=T-\tau_2}^{T} \gamma(t))^2. \quad (8)$$

$L(x)$ is the MSE distance between the original inputs $x$ and the averaged denoising predictions by the DAE (averaged over $\tau_2$ time steps). $T$ is the simulation length and we select the last segment of length $\tau_2$ where it is more possible to reach a convergent state (Fig.2c). $Z$ is a normalization factor.

Notably, the objective of the training process is consistent with the constructive nature of the perception and only related to multi-object inputs as well as the firing rate of neurons. Therefore, the grouping of objects and the neuronal synchrony are not explicitly guided by the objective function itself. Instead, they arise "effortlessly" as an emergent phenomenon along with anticipating the world[32], like an infant[4].

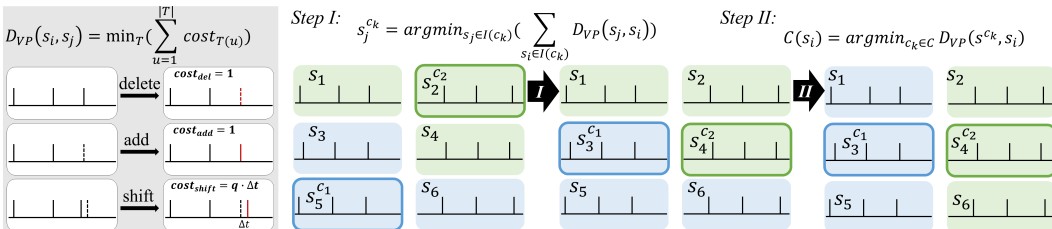

Figure 3: The illustrative scheme of SynClu. Gray box illustrates how VP metric($D_{VP}$) is computed. $s_{i/j}$ is the spike train of neuron $i/j$. $T$ is an arbitrary list of transformation to transform $s_i$ to $s_j$. $T(u) \in \{$del,add,shift$\}$. $D_{VP}$ is defined as the minimal cost to transform one spike train $s_i$ to another $s_j$. The right side shows how SynClu cluster the neurons (exemplified for a single clustering iteration). Step I: from clustering assignment $I(c_k)$ to updated k cluster centers $s_i^{c_k}$. $I(c_k)$ is the set of spike trains in the cluster $c_k$. Step II: from k clustering centers to updated cluster assignment $C(s_i)$.

### 3.5   Gredients and unrolling

The second challenge is how to train the GUST in a gradient-based manner, given that the end-to-end training of the GUST is not naturally compatible with the standard deep learning framework due to the binary and dynamical nature of the SCS.

The challenge comes from two parts. One is the non-differential elements in SCS, like spiking non-linearity (H, eq.4), coincidence detector (CD, eq.6) and refractoriness ($\delta_i$, eq.5). Another is the long simulation period ($> 100$), which makes it harder to backpropagate the gradients due to stability issues [39]. For the spiking non-linearity we approximate the Heavistep function H as a piece-wise linear function to compute a surrogate gradient[40, 41]. The same strategy is used to pass the gradient through the threshold of coincidence detectors. As for refractoriness, we cut off the gradient through the refractory variable $\delta_i$ to simplify the computation graph. As a result, the refractory variable acts as a gating variable so that the gradients can only be passed through neurons that are not in their refractory periods in the unrolling graph (green dots in the SCS in Fig.2c). Lastly, the transmission delay naturally shortcuts the depth of the backpropagation for $d$ times so that the backpropagation of errors along the simulation period is much more efficient (green lines in Fig.2c). The green lines and green dots in Fig.2 (c) stand for the shortcuts or neurons where the gradients can be back-propagated.

### 3.6   Evaluation

The last challenge is how to quantitatively measure the neuronal coherence in a high dimensional spike coding space and evaluate the grouping performance based on neuronal synchrony.

The evaluation of the coherent neuronal groups requires a clustering method directly based on the temporal structure of the spike firings. Therefore, we develop SynClu (Fig.3), a clustering method that combines a spike-timing-based metric and K-medoids clustering together. The Victor-Purpura metric ($D_{VP}$) that we use is a classical non-Euclidean metric for analyzing the precise temporal coding in the visual cortex[33]. The K-medoids[42] thereby iteratively assign K clustering centers($s_i^c$) among given neurons ($s_i$) to minimize the inner-cluster VP distance. The convergent clustering assignment is compared with the ground truth grouping by the adjusted mutual information (AMI)[43] to evaluate the grouping quality (1 is perfect grouping while 0 is the chance level). The Silhouette score[44] indicates the inner-cluster coherence level of the spiking pattern, therefore is denoted as the synchrony score (SynScore).

## 4   Experiments and results

Since the grouping is formalized as a pixel-level segmentation of visual scenes composed of multiple objects (Section.2 Grouping), synthetic images composed of multiple Shapes[45] are generated for evaluation. By generating a different number of shapes (Fig.1bottom, Section.2 Combinatorial generalization by grouping), we can control the statistical structure of the training data and testing data, as well as access to the ground truth of grouping assignment. This allows us to evaluate the grouping performance and further evaluate the combinatorial generalization ability of the approach.

In the following sections, we unfold the grouping property of GUST step by step. We first demonstrate the emergence of neuronal coherence during the simulation of a well-trained GUST, which softly indicates the grouping assignments (Section.4.1). To figure out why the intriguing grouping capability is learned without explicit supervision, we dive into the details of the training process. Notably, it is shown that the DAE of GUST gradually learns to denoise single objects instead of multiple objects (Section.4.2). Further visualization of the output attention map $\gamma$ shows that the DAE learns in a phasic manner, from coarsely predicting the global features (position) to additionally capturing local features (shapes) of single objects, which may provide a heuristic explanation of the mechanism. Lastly, given that GUST can learn to segregate multi-object scenes into grouped single-object representations during training, we study whether the single-object representation can be composed to understand novel scenes with more or fewer objects than ever encountered during training (Section.4.3). In other words, we ask whether the GUST generalizes its grouping mechanism in a combinatorial manner. For visualization and analysis in diverse cases, see Appendix A.9 $\sim$ A.11.

## 4.1 Emergence of neuronal coherence for grouping

We train the GUST on 54000 images composed of three random generated shapes (squares / up-triangle / down-triangle) located at different locations. We add salt&pepper noise to the input and train the GUST to denoise the image by its averaged firing rate during the convergent phase (last $\tau_2$ steps). In this section, we show the simulation result of the trained GUST when perceiving a randomly selected image. The GUST is simulated 3-times longer than training to confirm the stability of the grouping. The SynScore and AMI is evaluated every $\tau_2$ steps based on latest spiking patterns of length $\tau_2$ during the simulation. $\tau_2 = 10$.

The Synscore in Fig.4 gradually converge to 0.9 from 0, showing that the neurons in SCS evolve into a coherent state from random firing. Besides, the similar convergent curve of the AMI score further shows that the neuronal coherence successfully reflects the grouping assignment.

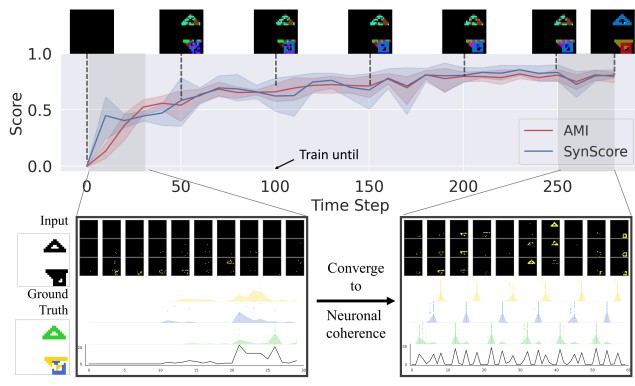

Figure 4: The simulation of a well-trained GUST on a randomly selected image. Top: coloring results based on the temporal structure of the spiking patterns. Middle: SynScore and AMI along the simulation, evaluated every 10 time steps, averaged over 5 random seeds. Bottom (in the black box): zoomed in spiking patterns during the initial 30 steps and last 30 steps. Upper pattern: snapshot of firing neurons at each time step (from left to right, from up to bottom); Middle pattern: spiking plot of neurons (similar to Fig.1b). The coloring is based on the ground truth grouping. Lower pattern (the back curve): the averaged population activity (LFP, similar to Fig.1b).

From the visualization result (Fig.4 top,bottom), it is more evident that in the convergent phase neurons related to the same object fires synchronously within a very short time window ($\tau_1 = 2$) while neurons relevant to different objects fires out of phase. Therefore, the neurons and features are grouped in the time dimension softly, which means that features are not forced to fit into a discrete slot but seat softly in the neighbourhood of a gamma-like temporal coordinate (Fig.1b, Fig.4black curve of LFP in the convergent phase)[46]. Notably, the temporal coordinates are not predefined as in slot models but self-emerged as an emergent phenomenon during the simulation, providing the potential to group objects more flexibly and dynamically.

However, why GUST can learn to group single objects in the first place? It seems intriguing because the GUST is only trained to denoise the multi-object images so that no information about single objects or grouping is leaked to GUST during training. To unravel the mystery, we dive into details into the training process in the next section.

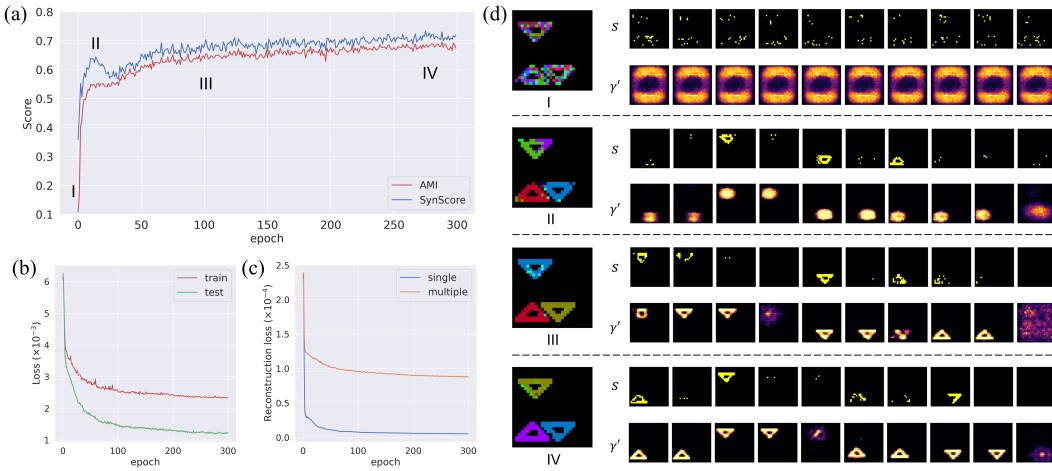

Figure 5: (a) AMI score and SynScore during the training epochs. (b): training / testing loss; (c) DAE reconstruction loss of single / multiple objects during the training process. Note the reconstruction losses of single / multiple objects have a ten-fold difference. (d) Visualization of SCS representation and DAE feedback along training epochs. From top to bottom: 4 examples selected in the 0/10/100/300 epoch during the training process, marked as I/II/III/IV in (a). Left: coloring of SCS neurons based on their spike firing time in the convergent phase. Right: spike firing sequence ($s(t)$) and feedback sequence ($\gamma'(t)$) (in the same convergent phase). In epoch 10 (marked as II), the feedback is a vague attention map while in epoch 100/300, the feedback is the precise reconstruction.

## 4.2 Learning to group without supervision

The AMI score and SynScore in Fig.5a is evaluated at the end of each epoch during the training, averaged over all testing images. The scores are based on the convergent phase in the simulation. Simulation in testing is 3-times longer than that during training. Surprisingly, even though the GUST is only trained to reconstruct the superimposed input by the averaged firing rate (eq.8), it seems to gradually discover a way to segregate the scene into a self-organized temporal structure and represent the building blocks by the neuronal coherence, which is illustrated by the simultaneously increasing AMI and SynScore along the training epochs (Fig.5a).

As expected, the denoising loss is decreased during the training (Fig.5b). Notably, it seems that the 3-times-longer testing process further improve the denoising performance since the denosing loss in the testing phase is much lower than that of the training phase (Fig.5b). Therefore, the GUST can stably generalize when using much longer iteration steps at test time.

To figure out what the DAE learns to reconstruct, we evaluate the reconstruction loss of the isolated DAE separated from the GUST architecture during the training (Fig.5c). The reconstruction loss is evaluated as the MSE loss between the image inputs and the denoising outputs of the DAE. Two types of images are used. One only contains a single randomly generated shape and the other contains three randomly generated shapes as used in training and testing. Surprisingly, the performance of reconstructing a single object improves with the training process (Fig.5c, single reconstruction loss). On the contrary, the reconstruction of multiple-object is not increased as saliently (ten-fold difference!). Denoising single objects by DAE is an important support to group single objects from multi-object images[37] (Also see Appendix). However, given that no explicit information about individual objects is provided to the DAE, how does it achieve the identification of a single object at all?

To this end, we further visualize the spike representation in the SCS ($s(t)$) and the top-down attention given by the DAE ($\gamma'(t)$).

It can be seen that the DAE of GUST learns in a phasic manner (Fig.5d I to IV). At the beginning of the training process, it first learns to capture the global feature of the object (position) and feeds back coarse attention maps focusing on local regions covering each single object ($\gamma'(t)$ in Fig.5d II). Despite of the vague feedback, the gating effect of input (eq.2) further refine the firing pattern so that

each clear-cut shape fires in synchrony ($s(t)$ in Fig.5d II) because irrelevant neurons in the attended local regions are kept from firing by zero driving signals. However, if objects come closer or even overlap, such rough attention mechanism may lead to incorrect groupings.

In the second stage, the DAE learns to further capture the local shape information to provide more precise denoising reconstructions (Fig.5d III,IV). Notably, the higher-quality feedback also improves the neuronal coherence and grouping performance in SCS (Fig.5d II to IV).

How can we understand the phasic learning phenomenon? Here we provide a heuristic explanation which highlights the role of the biological constraints. Although the learning target is the superimposed reconstruction by the averaged firing rate, the DAE needs to extract "adequate" information for top-down prediction. However, the biological constraint of limited temporal window CD (eq.6) makes it difficult at the beginning because neurons fire randomly (eq.3) and sparsely (eq.5) so that only a small part of firing can be observed at any temporal window (Fig.5d I). Thus, to achieve the objective with such limited observations, the DAE chooses to learn a detour, which is easier: Inferring a local location based on very sparse firings. This is the attention map in the first phase. However, this unexpected by-product has profound influence on the subsequent learning process: the local attention map breaks the symmetry of the neuronal responses to the input and the refractory dynamics(eq.5) thereby separate neuronal groups at alternative timings, albeit with lower precision. Thus, the original learning objective is "decomposed" into reconstructing each observed single object in the following training epochs. In turn, more precise top-down inference modulates the spike timing into a more structured synchrony. The ablation of biological constraints are shown in Fig.6a and detailed discussion of their role in grouping can be found in Appendix A.3.2.

### 4.3 Combinatorial generaization to novel scenes

Finally, we can ask: Does the GUST generalize the grouping mechanism in a systematical way? Specifically, does it generalize to novel scenes with more or fewer objects than ever encountered during training? For this purpose, we alter the number of objects both in the training data and testing data. The AMI results are shown in Fig.6b,c,d. For comparison, we compare the grouping performance with a similar benchmark model RC-bind[47] and Slot Attention[48], that also utilize autoencoder architectures to achieve grouping. Background is considered when computing the AMI.

The AMI results show that the GUST generalizes to scenes of varied number of objects (more than 0.5 in all cases and achieve 0.7 in 4 cases), outperforming the RC-bind (around 0.3) and slot-attention (failure). This indicates that the learned symbol-like entities can retain themselves and be composed according to the combinatorial nature of a novel scene.

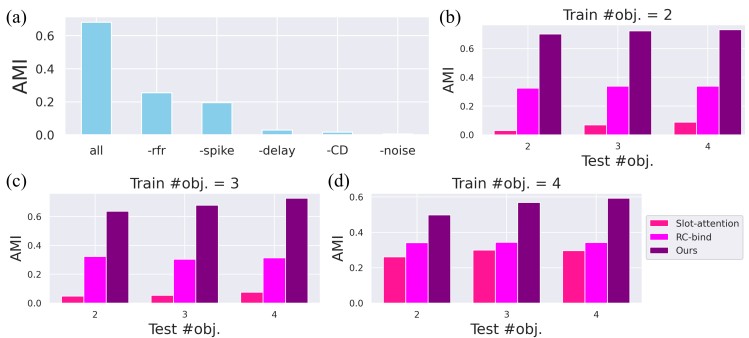

Figure 6: (a) Ablation of biological constraints. Compared with original GUST (all), removing stochasticity (-noise), refractoriness (-rfr), feedback delay (-delay), coincidence detectors (-CD), spiking non-linearity (-spike) will all harm the grouping performance (AMI) and even cause the failure of the grouping. (b)(c)(d) AMI score of GUST, RC-bind, and Slot Attention, trained / tested with 2/3/4 objects.

## 5 Related work

**Slot grouping.** There is a broad range of research for object grouping in the artificial neural network[47, 14, 49, 50, 51, 52]. Most works use a particular representation format, the slot. These representational slots are predefined into the network architecture to explicitly separate the objects with fixed capability. On the contrary, the temporal grouping in GUST assigns each group implicitly as an emergent property of the network dynamics. Without explicit human knowledge, the grouping is more general and flexible.

**Complex-value grouping.** Grouping based on complex-value approach[53] abstracts away the firing rate and firing timing into the amplitude and phase of the complex number, and readout the grouping information by specific complex-value activation functions. The approach has the desirable property of soft grouping. However, due to the limited range of the phase value, the number of objects is limited for grouping[54]. In contrast, the range of temporal dimension is potentially unlimited.

**DASBE**[37] is a recently developed hybrid model[55] to bind features through spike timing synchrony and generative attention. However, DASBE requires the isolated training of the DAE to denoise single objects. In contrast, instead of showing the existence of binding solution by construction, GUST shows that the grouping can even be (1) learned (2) in fully unsupervised way (3) in a single neural network directly perceiving (4) multiple objects through the (5) structural bias of biological constrains and the learned blocks can (6) generalize combinatorially.

**Binding problem in neuroscience**. How to bind distributed features in the neural system is one of the fundamental questions in neuroscience and cognitive literature. Towards the binding problem, two mainstream theories have been proposed. On the one hand, in the temporal binding theory (TBT)[17], it is reported that the temporal coherence states are related to the binding of features and is highly related to the constructive nature of the top-down feedback in a processing hierarchy[32]. On the other hand, feature integration theory (FIT)[56] is based on attention mechanism. It assumes a two-stage processing. In the first phase, features are extracted into a location map. In the second phase, a top-down attention "spotlight" searches the location map to group the features. It is still open which theory captures the nature of binding in the brain[57] and it is likely that brain uses both strategies[7]. Since the GUST minimally integrates both neuronal synchrony and attention mechanism in a closed way, it is promising that GUST can also serve as a unified data-driven modeling framework for bridging both TBT and FIT theories to explain the binding process in neural circuits in the future.

# 6    Conclusion

In this paper, we develop GUST, a brain-inspired unsupervised grouping system that dynamically finds a coherent grouping solution by the iterative processing between spiking dynamics and denoising feedback. The grouping is learned with a succinct denoising loss and can be generalized to represent scenes of combinatorially different structures.

# 7    Acknowledgements

This work was supported by the National Key Research and Development Program of China (no. 2021ZD0200300), the National Nature Science Foundation of China (nos. 62088102, 61836004).

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
