# GUST: Combinatorial Generalization by Unsupervised Grouping with Neuronal Coherence

## Contents

# A  Appendix

In Section A.1, we highlight some limitations of our work as well as potential directions for future works. In Section A.2, we discuss possible positive and negative broader impact of the GUST. In Section A.3, we discuss the role of each biological constraint in detail and also provide an explanation of the grouping mechanism from an energy viewpoint. Based on the explanation, we further view the GUST from different levels in Section A.4. In Section A.5, we introduce the Shapes dataset (and its variations in the main text) and how they are generated. In Section A.6,we show the computing resources and details for training GUST. In Section A.7, we list the hyper-parameters of GUST. In Section A.8, we provides more detailed explanations and additional results to supplement the experiment figures in the main text. In Section A.9, we show the grouping result of GUST on an image of low quality and discuss the softness of grouping. In Section A.10, we show the multi-solution property of the GUST(mentioned in the main text). In Section A.11, we provide more analysis on how GUST work in more complex cases (eg. objects of different sizes, of different color; in front of novel unseen objects, MNIST objects, moving object[1], etc). In Section A.12, we introduce the metrics or measures we use in the paper including AMI score, SynCore, and Victor-Purpura metric.

## A.1  Limitations and future works

In this section, we highlight several limitations that could be addressed in future works.

**Binary input**. The inputs for GUST are the binary images which are compatible with the binary coding of spikes, but it is desirable to relax such constraints to real-valued images. Generally speaking, the brain faces the same issue, which mostly relies on binary spiking neurons to represent entities potentially of real-valued features. Therefore, it is promising to find a brain-inspired solution. On the one hand, a conversion (or encoding strategy) from real-valued input to binary code may resolve this problem[1, 2]. More specifically, different colors are represented by a group of binary neurons in cortex with receptive fields in a restricted wavelength range[3]. In other word, brain uses binary assembly code (similar to a kind of flexible one-hot encoding) to represent the color space. For example, red neuron, deep red neuron, blue neuron, light blre neuron, green neuron, etc. In this way, the real-valued color space are discretized / divided into zones, each of which are represented by a different group of binary neurons. The precision of the color encoding depends on how fine-grain the color space is divided, which might further depends on the scale of the cerebral cortex[3]. While the whole RGB space may be too large to divided it into very fine-grained zones, we could also find inspiration from brain to reduce the complexity. More specifically, different color are represented with different sensitivity or efficiencies due to adaptation during evolution, which is summarized as fundamental chromaticity diagram (CIE)[4, 5, 6]. Such coding scheme extremely reduce the representation complexity and could be exploited to deal with real-valued color images. Neuromorphic chips[7] that optimize for brain-inspired computation further reduce the complexity. In Section A.11, we provide a primary simulation result on colored Shapes, following the aforemenioned coding strategy, to show that introducing color does not bring fundamental harm to the dynamics and learning of GUST. More detailed study of GUST to deal with color images following CIE coding scheme leaves to future work.

On the other hand, as explained in the following, grouping in latent space instead of visible space (at pixel level) may more naturally resolve this problem in the future. More specifically, the input to the GUST is not restricted to the original pixel-level image, but could be the output of an up-stream

---

[1]Moving objects may induce a fundamental issue for temporal binding: the dual use of time. It is discussed also in Section.A.11

encoding module like CNN[8], which is a set of more abstract feature maps of reduced dimensions. Then we could generally binarize the reduced feature map before fed into the GUST. In other words, the grouping occurs in latent space or intermediate space. Similar strategy has also been discussed in the design of GLOM by Hinton[9].

**2-D vs 3-D**. Only 2-D objects are considered in this work, partially due to the binary nature of the inputs. In future work, it is preferable to study the binding of 3-D objects which are more realistic, for example, CLEVER datasets[10].

**Void background**. The background is set to be 0 for simplicity, however, it is desirable to consider background with noisy patterns or with irrelevant objects in the future, which is also more realistic and challenging.

**Pixel-level grouping**. In this work, grouping is studied in visible space (pixel-level). The advantage of pixel level grouping is that the representation is interpretable and the ground-truth of grouping assignment is available. However, it is desirable to develop grouping in (binary) latent spaces of the deep neural network, so that higher-level or even hierarchical-level features can be grouped. With this extension, the constraint of binary inputs can be relaxed because real-value input can be transformed into binary latent space by a trainable encoder. Since the abstract latent space is usually much more factorized and of much reduced dimension (more like the structure of visible space in this paper), it provide ways to generalize the method into more complex inputs such as real images. We leave the 'grouping in latent space' as future works. However, the challenge is that representation in latent space is harder to be interpreted and the grouping quality is harder to be evaluated. A decoding method may solve this challenge in the future[11].

**Evaluation**. In this work, the evaluation is based on ground-truth assignments (K is needed for K-Medoids clustering). However, since the grouping process is in principle of multiple possibilities, such ground-truth can not impartially evaluate the grouping quality (only provide partial measures of the grouping quality). The desirable evaluation of grouping that is independent of prior knowledge of ground-truth is still an on-going challenge for object discovery field and may need further exploration.

**Applying GUST for computer vision field**. In this work, we focus the discussion of the grouping on synthetic dataset, leaving it uncertain how current GUST could be directly used to deal with real-images or videos in deep learning field. First of all, dealing with real images is a common short-coming of the object-centric learning field, which put more strength to deal with synthetic data[12, 13, 14, 15, 16, 17, 18, 19, 17, 20], partly due to the interpretable and theoretical nature of this field: object-centric representation need to be interpretable and the mechanism should be general. Secondly, the original motivation of this work is to build-up the concept that grouping in time could be learned end-to-end from superimposed stimuli in a human-like unsupervised manner, with the help of biological constraints as inductive bias. The learned symbol-like representation can generalize combinatorially. Following this motivation, we keep the model architecture (delayed top-down / bottom-up between spiking neuron and DAE), model implementation (only simple DAE are used), loss function (succinct MSE loss), minimal. Therefore, it is better to regard GUST as a basic prototype of a list of extensions, when augmented with more delicate front-end/back-end modules, more complicated network structure and dataflow, more loss terms (regularization) and data augmentation. Since the mechanism illustrated in the main text is general, compatible with a wide range of ANN models, thus it is promising to develop GUST applications without fundamental restrictions. Interestingly, there is a similar research line in object-centric learning field, from simple binary objects[17, 18] to more complex scenes[19, 15](but still mostly restricted to synthetic data, yet of higher complexity), which could be compared with the stage of the current work. As [17], realizing the unsupervised learning of temporal grouping should be an important step to trigger the following models, like[19, 15], in deep learning field. Therefore, it might be helpful to keep the model minimal at the preliminary stage. Future work is desirable to extend the basic GUST architecture to deal with more complex situations, some preliminary case studies are shown in Section.A.11.

**GUST for neuroscientific modeling**. In this work, we are motivated from a computational viewpoint: the unsupervised grouping for combinatorial generalization in vision. However, how to bind distributed features in the neural system is also one of the fundamental questions in neuroscience and cognitive literature, also related to Gelstalt Psychology[21]. It is reported that the temporal coherence state is related to the binding of features and is highly related to the constructive nature of the top-down feedback in a processing hierarchy[22]. Therefore, a more ambiguous future work of GUST is to develop a data-driven modeling framework for explaining the binding process in neural

circuits, integrating the temporal binding theory and the attention-based feature integration theory (FIT)[23].

**Limitation from neuroscience view point**. The GUST is a minimal model of explaining binding, rhythm, synchrony and attention. However, as expected, it is not a detailed model of neural circuits. Here we list some of the ingredients that are not reflected in the current but worth being considered in the future. First, short-term plasticity and recurrent dynamics within the SCS are not considered in GUST, this makes the representation in SCS highly dependent on the persistent presence of external stimuli. Once the stimulus is off, the spiking activity stops. However, it is likely that it is not a fundamental limitation of the model and could be resolved in the future. Second, a broad range combinatorial code is present in the brain, besides synchrony code. For example, polychrony is argued to be more common in the brain than synchrony and is more powerful to solve combinatorial binding[24]. Third, balance of excitation and inhibition is not considered in the model. Actually, the inhibitory effect is only considered neuron-wise, as refractory effect. More generally, such inhibitory effect should be implemented as groups of inhibitory neurons.

## A.2 Broader impact

On the positive side, the GUST groups object with neuronal coherence in the spike coding space in an unsupervised manner. The mechanism by principle is not limited to certain modality or certain feature levels. Thus, it may help develop brain-inspired perception systems. Besides, with biological relevant features (eg. delayed coupling) and phenomenon (eg. synchrony), GUST may also act as a data-driven biological model to understand the perception process in the brain.

On the negative side, since the GUST is an unsupervised system, it is harder to control / evaluate what the GUST learns. The current GUST is only trained on simple synthetic dataset and learns to group at superficial pixel level, therefore the representation is highly explainable. However, grouping in latent space on real-world dataset requires to develop evaluation and visualization method to make the representation in latent space more understandable. We believe this may serve as a step toward more transparent and interpretable predictions

## A.3 Biological Constraints

In this section, we explain the the roles of biological constraints in detail. The eq.1 $\sim$ eq.7 in the main text (also copied below) conresponds to seven biological constraints in the cortical circuits. To be more specific:

Transmission delay[25] ($d$):

$$\gamma_i'(t) = \gamma_i(t-d).$$ (1)

Gating effects ($\cdot$) of driving and modulatory signals [26, 27, 28, 29]:

$$\rho_i(t) = \gamma_i'(t) \cdot \tilde{x}_i.$$ (2)

Stochasticity ($\epsilon$)[30]:

$$s_i(t) = H(\rho_i(t) + \epsilon; \delta_i(t)).$$ (3)

Non-linear spiking activation ($H$)[30]:

$$H(u_i; \delta_i) = \begin{cases} 1, & \delta_i = 0 \land u_i \geq 1. \\ 0, & \delta_i = 0 \land u_i < 1. \\ 0, & \delta_i > 0. \end{cases}$$ (4)

Refractoriness ($\delta$)[30]:

$$\delta_i(t) = \begin{cases} 0, & s_i(t) = 0 \land \delta_i(t-1) = 0. \\ \delta_i(t-1) - 1, & s_i(t) = 0 \land \delta_i(t-1) > 0. \\ \delta, & s_i(t) = 1. \end{cases}$$ (5)

Coincidence detector ($CD$)[31]:

$$s'_i(t) = CD(s_i(t), s_i(t-1))$$
$$= \begin{cases} 1, & s_i(t) + s_i(t-1) \geq 1. \\ 0, & s_i(t) + s_i(t-1) < 1. \end{cases} \tag{6}$$

A processing hierarchy of constructive nature ($f, g$)[32, 22, 33]:

$$\gamma_i(t) = g(f(s'_i(t))). \tag{7}$$

How does these biological constraints shape the dynamics in the GUST so that an iterative temporal grouping state emerges during the simulation? We provide an heuristic explanation from an energy perspective.

### A.3.1 Energy perspective

In this part, we provide intuitions of how the biological constraints can act as inductive bias to enable temporal grouping in an energy perspective. The biological constraints are added incrementally to build the whole system step by step. First, only the hierarchical processing ($f$,$g$), stochasticity ($\epsilon$), spiking non-linearity ($H$) are considered and other factors are ignored. In this case, the system is a Markov chain of $s$ (eq.3) with transition matrix parameterized by $f$ and $g$. Thus, under the general condition of irreducibility and aperiodicity, an invariant distribution $P(s)$ exists[34]. Thus, we can define $E(s) = -log(P(s))$ as the energy landscape. And the dynamics of $s$ amounts to searching for the global minimum of $E(s)$. If $f$ and $g$ are the encoder and decoder of a DAE denoising "single" objects (denoted as $s^k$ for object $k$), respectively, then $P(s^k)$ increases and thus $E(s^k)$ is lower. For example, the red, green and blue columns in the Fig.1a may stand for three low-energy states where a single object is represented.

Second, if we add the modulatory constraint that uses input as a multiplicative gate, the landscape is then further shaped by the external input, reducing the searching space. For example, states that are inconsistent with the input (eg. objects not contained in the image) get elevated (Fig.1b, yellow columns). Under the constraint of input, even the imprecise top-down feedback by DAE is possible to search a desirable state (Fig.7 II in the main text). Third, considering the delay ($d$) constraint, it amounts to create d independent copies of the energy landscapes, each describing an independent sampling process (Fig.1c). For example, if $d = 2$, then the states at the odd number of time steps (1/3/5/7...) become independent of the states at the even number of time steps (2/4/6/8...).

Lastly, refractoriness ($\delta$) and temporal integrative window (CD) link the adjacent landscape copies so that the latter energy landscape reshapes itself according to sampled states at the previous step (marked by the stars in Fig.1 (d)~(f)). This dependency elevates the states that have been previously sampled (Fig.1 (d)~(f), red and green arrows). This encourages the traversal of possible object states in these d copies. Some of the d copies correspond to the temporal coordinate ($z_i$) mentioned in Fig.1 in the main text.

In sum, these biological constraints together can be seen as inductive biases to bias GUST to find a grouping sollution with neuronal coherence. Note that the picture is based on the premise that the DAE can learn to denoise single object. A similar proof in a point of view of the attractor dynamics can be found in [16]. The ablation study in Fig.8 in the main text further shows the indispensable roles of these biological constraints (eq.1~eq.7).

### A.3.2 The role in the phasic learning process

The discussion in the Section.A.3.1 assume that the DAE can denoise single objects, which is indeed the case in a well-trained GUST (Fig.6 and Fig.7 in the main text). But how it can be achieve gradually during the training process? That is the question in this Section.

As explained in the main text Section 4.2, the limited time window of CD, the gating effect of inputs and the refractoriness may bias the DAE to learn to denoise single objects. Firstly, refractoriness constrain the firing rate of each neuron so that the initial random firing is very sparse. Secondly, such sparsity makes it difficult to simply reconstruct the original input pattern with limited observation during the narrow temporal windows of CD. Thus the DAE learns to infer an imprecise local region that cover the objects, since it is easier. For example, if there are only 3 spike firing at a given time

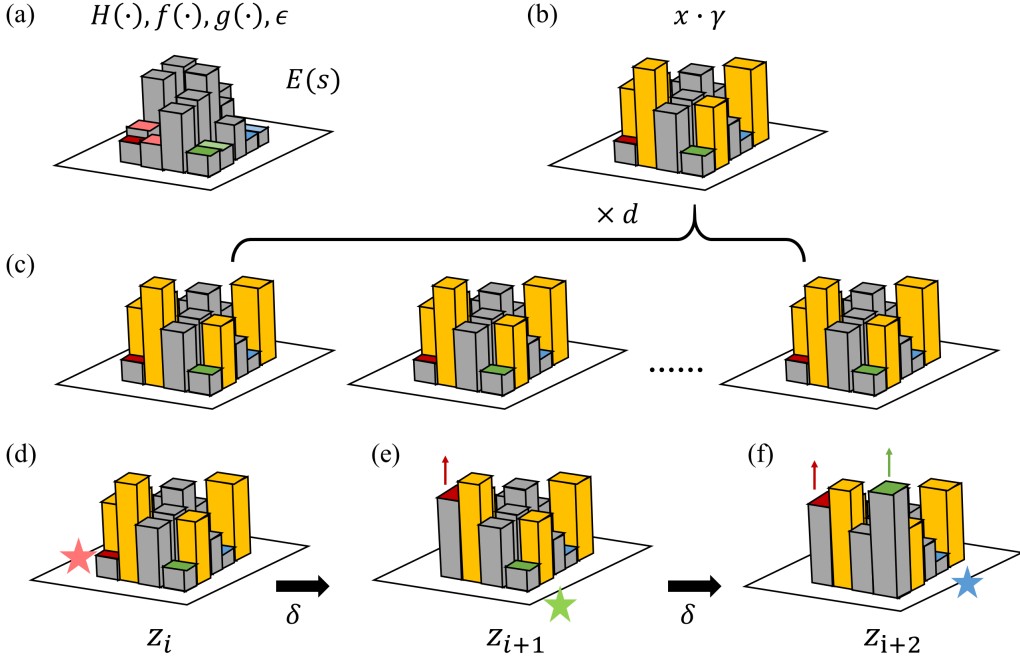

Figure 1: The energy perspective of the GUST. The energy landscape of states ($E(s)$) is indicated by bars of different heights. Each bar corresponds to the energy of a specific state (s) in the high-dimensional state space. The state space is composed of the collection of all possible spike firing configurations (of $28 \times 28$ dimension). For example, the spiking representation of a single up-triangle located at the center of the image can be considered as such a state ($s^k$) and a specific random firing pattern can be considered as another state. (a) The energy landscape when only considering $H(\cdot), f(\cdot), g(\cdot), \epsilon$. In this case, several local minimum may exist, indicated by red, green and blue columns. These states are more likely to occur under the intrinsic dynamics of the system. (b) If the gating effect of the external inputs are added, states that are not consistent with the inputs are excluded (indicated by elevated columns in yellow). This is because although likely to fire under the intrinsic dynamics, irrelevant neurons are excluded from firing due to the binary gating of the external input. This gating effect refine the patterns, illustrated by the fewer red/green/blue columns in (b). (c) If the transmission delay $d$ is further considered, it amounts to create d independent copies of the system in (b), each evolving on a separate delay path. For example, if $d = 2$, then states at 1/3/5/7...time steps is independent of states at 2/4/6/8... (d)~(f) If the refractoriness $\delta$ is further added, the copies in (c) become no longer independent. More specifically, if the present state is the red column (d, red star), then the the red column is elevated at the next time step due to the refractory period, so that the next state may alternate to the green column (e, green star). (f) Similarly, the following state may alternate to the blue state (blue star) since both the red and green states are elevated due to the refractory period. These alternative local minimum states act as temporal coordinates to group the object related features ($z_i, z_{i+1}, z_{i+2}$ in (d)~(f)).

step (very sparse), the DAE can still infer a region that contain the most spikes (note that 3 spikes are not sufficient to infer one or multiple precise Shapes in most cases). Thirdly, the gating effect of inputs refines the synchrony patterns so that such imprecise top-down feedback is sufficient to create much more precise synchronous neuronal groups. This by-product of learning collects enough spike firings within the limited observation window of CD during the following simulation period, so that the DAE can learn to reconstruct each single obeject more precisely.

In sum, The biological constraints both bias a temporal grouping solution and bias the learning of DAE to denoise single objects.

### A.4 Three levels

Interestingly, in a point of view of David Marr's three-level framework[35, 36] (Fig.2), the GUST aims to achieve combinatorial generalzation (computational level) through grouping in time (algorithmic level). Such grouping ability is biased by the 7 biological constraints (implementation level).

Based on the binding framework proposed in [20], the computational level (generalization ability) and the algorithmic level (temporal grouping mechanism) are linked by representing objects in the time dimension (representation), learning to segregate scenes into single object representations during the training (segregation) and then composing the learned single object representation to represent scenes of varied number of objects during the testing (composition). The algorithmic level and implementational level are linked by the explanation in Section.A.3, which highlights the role of biological constraints as inductive bias for learning.

On the one hand, such connection among levels is helpful to figure out which biological ingredients are computational useful and could be integrated into ANNs literature. On the other hand, such connection has the potential to develop GUST as a data-driven cortical circuit model to understand human vision in a systematical way[37].

### A.5 Dataset

The dataset we use is the synthetic images composed of multiple Shapes (Fig.3)[38, 18]. Three possible shapes (square/up-triangle/down-triangle) is randomly selected. The position of each shapes is randomly generated in a $28 \times 28$ image. The number of objects can be flexibly controlled for generating datasets of different object number. In the experiment of combinatorial generalization, datasets containing images composed of 2/3/4 objects are generated. The statistical property of the dataset is shown in Fig.4. The input to the GUST is added with salt&pepper noise during the training process ($p = 0.65$, See Table.2).

### A.6 Training details

#### A.6.1 Resources

Our experiments have been performed on ubuntu 16.04.12 with devices: CPU (Intel(R) Xeon(R) CPU E5-2640 v4 @ 2.4GHz) and 4×GeForce RTX 2080 Ti. The python version is 3.6.3.

#### A.6.2 Network architecture and training hyperparameters

The details of training neural networks are shown in Table 1. The learning rate decays with the training epochs. For details, the learning rate is reduced by half in 60, 100, 200 epochs (300 epochs in total).

#### A.6.3 Loss function

The loss function for training GUST on the Shapes dataset is (copied from the main text):

$$L(x) = (x - \frac{1}{Z} \sum_{t=T-\tau_2}^{T} \gamma(t))^2 \tag{8}$$

**The time window $\tau_2$ and normalization $Z$.**

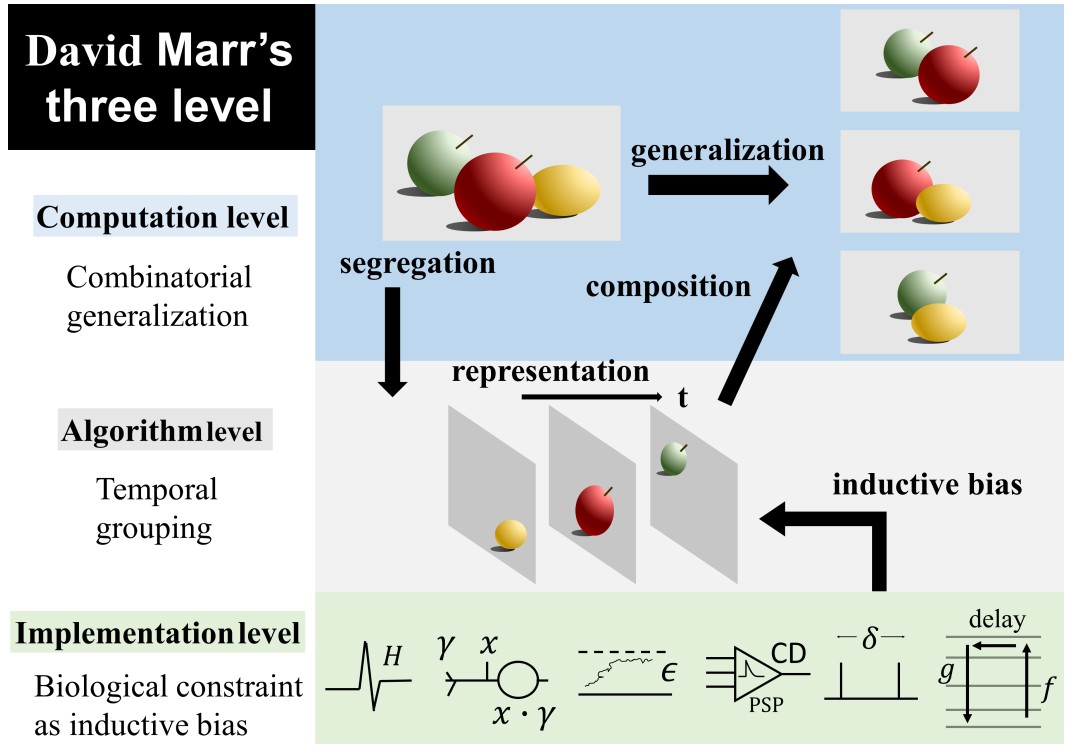

Figure 2: Viewing the GUST from the David Marr three-level framework. The compositional generalization is enabled by segregating raw input into building blocks that are represented in the temporal dimension and further composing the learned building blocks to understand novel scenes. The temporal grouping ability is biased by 7 biological constraints: spiking non-linearity(H), modulatory gating($\cdot$), stochasticity($\epsilon$), coincidence detector(CD), refractoriness($\delta$), delayed feedback($d$) in a procesing hierarchy ($f, g$).

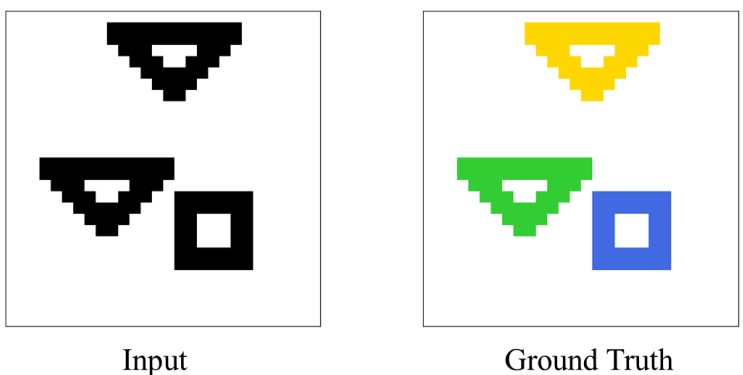

Figure 3: An example of Shapes dataset. Left: original image. Right: ground truth labeling.

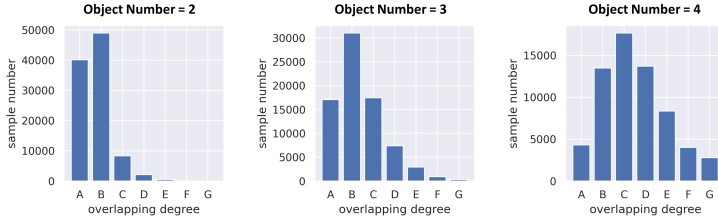

Figure 4: Overlap distribution in dataset of 2/3/4 objects (size = 60000). A: non-overlap; B: 1∼9 overlapped pixels; C: 10∼19 overlapped pixels; D: 20∼29 overlapped pixels; E: 30∼39 overlapped pixels; F: 40∼49 overlapped pixels; G: > 50 overlapped pixels. Each Square has 40 pixels and Each Triangle has 48 pixels in total. It is clear that (1) overlap is more common than non-overlap; (2) more object number causes higher overlap ratio; (3) Severe/sick overlap exists but is rare cases (E,F,G).

Table 1: Details of training neural networks.

| Name | Value |
|---|---|
| Encoder | [l]FC(784, 512) |
| ReLU() | |
| FC(512, 400) | |
| Sigmoid() | |
| Decoder | [l]FC(400, 512) |
| ReLU() | |
| FC(512, 784) | |
| Sigmoid() | |
| Simulation length | 100 |
| Learning rate | 0.01 |
| Minibatch size | 1024 |
| Salt&Pepper noise | 0.65 |
| Early stop | 20 epochs |
| Maximum epoch number | 300 epochs |
| Training method | Adam |
| Training dataset size | 18000 |
| Validation dataset size | 2000 |
| Test dataset size | 1000 |

The $\gamma(t)$ is computed based on the noisy inputs $\tilde{x}$ (add $x$ with salt&pepper noise, $p = 0.65$, See Table.1). The considered time interval is the last $\tau_2$ steps during the simulation: $[T - \tau_2, T]$, where T is the total simulation length. $\tau_2 = 10$ (Table.2). Since each spiking neuron is possible to fire more than once during the considered interval $[T - \tau_2, T]$, the term $\sum_{t=T-\tau_2}^{T} \gamma(t)$ needs to be normalized so as to be comparable to the binary input x. To this end, we regard both the input and the summed feedback term as an unnormalized probability distribution on the $28 \times 28$ pixel space. Therefore, we normalize both x and $\sum_{t=T-\tau_2}^{T} \gamma(t)$ so that the sum of both terms equals to 1. Since the normalized "distribution" is comparable now, we then compute the MSE difference between the two distributions. The $Z$ is an effective description of such normalization procedure.

**The noise and the unsupervised nature of the model.**

The general salt & pepper noise added to the input $x$ can be regarded as a generic internal noisy process inside the GUST (like Drop-Out[39]), which is also common in bio-systems, instead of a specific external data augmentation in self-supervised learning literature. Indeed, as explained in detail by Bengio in [40], 'denoising' is an unsupervised method (differ from augmenting data since it neither uses prior knowledge of images, nor produces extra labels). The same form of noise has also been used as loss for **fully** unsupervised grouping in Tagger[17]. Lastly, we have replaced DAE with VAE[41] so that only noise-free inputs ($x$) are needed (input noise is replaced by internal Gaussian noise in hidden layers) and it still achieves comparable AMI (**0.74**). In conclusion, the noise added to

Table 2: Hyperparameters of GUST in the experiments. $T_{train}$, $T_{test}$ and $T_{visual}$ is the simulation length in different cases. $\tau_2$ is used for computing the loss function (eq.8), evaluating AMI / SynScore, and coloring. $V_{th}$ is the threshold used in $H$ (eq.4) and $CD$ (eq.6). The latent size is the size of the latent space of the DAE in GUST. $p$ is the salt&pepper noise added to the input during training. During testing and visualization, the salt&pepper noise is removed. $K$ is the number of clusters for K-Medoids, always the number of object + 1(for background). $q$ is the timescale parameter in Victor-Purpura metric (Section A.12.1 and Fig.21).

| hyper-parameters | |
|---|---|
| $T_{train}$ | 100 |
| $T_{test}$ | 240 |
| $T_{visual}$ | 300 |
| delay | 10 |
| $\tau_2$ | 10 |
| $\delta$ | 9 |
| $V_{th}$ | 1 |
| $p$ | 0.65 |
| $K$ | #obj + 1 (bg) |
| $q$ | $\frac{1}{3}$ |

the input should be regarded as a flexible choice at implementation level, with various options, and should not harm the unsupervised nature of the model / framework.

**Understanding the loss function and its bio-plausibility.**

The loss function can be regarded as the MSE difference between the external driving input and the averaged "firing rate" of SCS neurons, since the feedback $\gamma(t)$ partially determine the firing rate of SCS neurons (eq.2). In the temporal binding theory, firing rate and firing timing encode feature content and grouping relation respectively[42]. Therefore, the loss function only guide the averaged feature content, leaving the grouping information totally unknown.

Besides, the loss function can also be considered as the difference between the correlation between the driving input and modulatory input, which may be related to the Hebbian plasticity between neurons in SCS and latent space[43]. Notably, the loss function does not include information about the spike timing structure to guide the neuronal coherence.

### A.7  Hyper-parameters

All the hyper-parameters used in the experiments are listed in Table.2. $T_{train}$ is the simulation length of training. $T_{test}$ is the simulation length of testing (computing the AMI score and SynScore in Fig.8,9,10,11 (also the test loss in Fig.6) in the main text). $T_{visual}$ is the simulation length for Fig.4 in the main text. $\tau_2$ is used for computing the loss function (eq.8), evaluating AMI / SynScore, and coloring (10 ($1 \times \tau_2$)). $V_{th}$ is the threshold used in $H$ (eq.4) and $CD$ (eq.6). The latent size is the size of the latent space of the DAE in GUST. $p$ is the salt&pepper noise added to the input during training. In other cases, the salt&pepper noise is removed. $K$ is the number of clusters for K-Medoids, always the number of object + 1 (for background). Same $K$ for RC-bind and SlotAttention (See Table.6 and Table.7). $q$ is the timescale parameter in Victor-Purpura metric (Section A.12.1 and Fig.21).

### A.8  Details for experiments

In this section, we provide detailed information of the experimental setting and additional results related to each Figure (Section A.8.1 $\sim$ Section A.8.5).

#### A.8.1  Details for Fig4

**Simulation length**

The simulation length in the training process is set to be 100 time steps ($T_{train}$ in Table.2) for more efficient error backpropagation (denoted by "train until in Fig.5"), while the length in visualization process is set to be 300 time steps to allow the system fully converge to a coherent state ($T_{visual}$ in

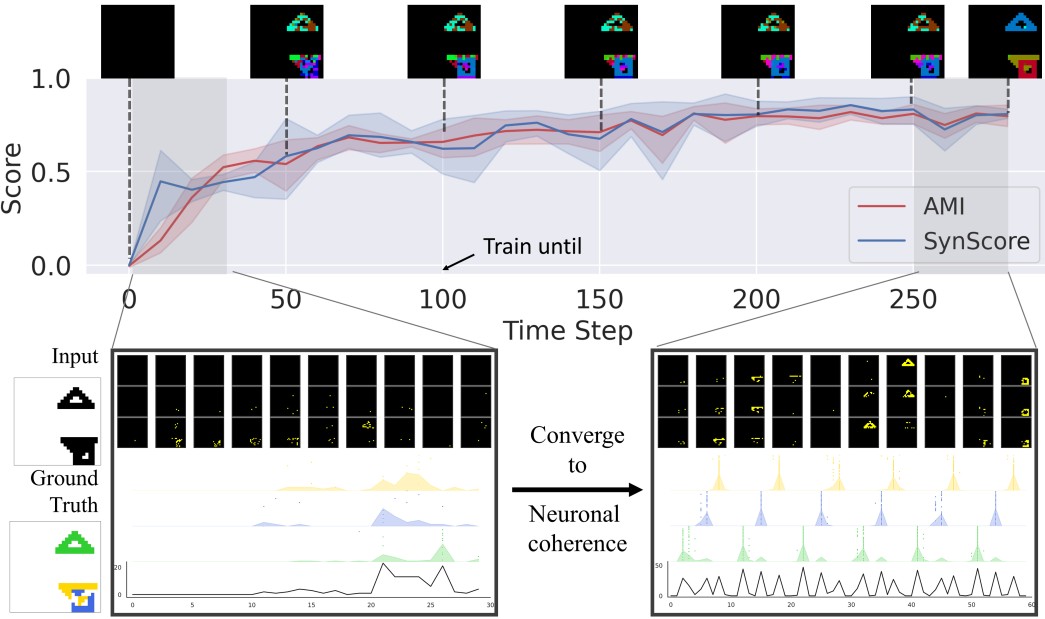

Figure 5: Copied from Fig.4 in the main text.

Table.2). Longer steps helps to confirm the stability of the grouping or the generalization of grouping with respect to longer simulation period.

There are three notable phenomenons during the simulation. First, the system converges fast in the initial 50 steps (achieve visible separation of objects in less than $5 \times \tau_2$ or 50 steps). Second, the grouping is stable even after the first 100 steps ($T_{train}$). Third, it is intriguing that both the AMI and Synchrony Score gets further improved after the first 100 steps ($T_{train}$). Such property indicate that the GUST can generalize its grouping ability with respect to the length of simulation.

**Salt&Pepper noise**

The Salt&Pepper Noise is added to the input during training ($p = 0.65$ in Table.2 or Table.1). During the simulation of the well-trained GUST, the noise is removed.

**Coloring**.

To visualize the temporal structure, we continuously assign an RGB color to each spike timing during the simulation. For example, if a neuron fires a spike at $t = 10$, then this neuron may be assigned as a red color. Thus, a temporal interval is needed to generate each coloring result.

More specifically, for the coloring results in the Fig4 (copied in Fig.5) of the main text, intervals of $\tau_2 = 10$ steps are considered for each coloring result (Table.2).

The color for each pixel is based on the corresponding spike firing time (by a mapping from the spiking phase to the RGB value). More details about the implementation can be found in the source code (file name: draw _color.py) (See main text for code availability)

**Scores**.

Since the SynScore and AMI Score are both based on a clustering process with respect to the "spike trains" of SCS neurons, intervals are needed to determine these "spike trains".

More specifically, intervals of $\tau_2 = 10$ time steps are considered for each SynScore and AMI Score (Table.2). And they are evaluated every 10 time steps during the whole simulation. In other world, we divide the whole simulation period into intervals of 10-time-step-length and compute the Scores based on each interval.

The Score are averaged based on five random seeds. Both score converge fast from 0 to more than 0.6 during the initial 70 steps and gets further improved to more than 0.9 after the first 100 steps ($T_{train}$).

**Spiking patterns in the black box**

The left box is the zoomed in visualization of the initial phase (first 30 steps, indicated by the left grey shadow in the Score plot) and the right box is the zoomed in visualization of the convergent phase (last 30 steps, indicated by the right grey shadow in the Score plot).

(Top) The patterns on the top (black background) is composed of 3 rows of shotsnaps of the SCS neurons (each row is composed of 10 shotsnaps). Thus, there are 30 snapshots in total (in each box). Each snapshot correspond to the spike firing pattern at a time step. More specifically, in the left box, the fist row is 1∼10 steps. The second row is 11∼20 steps. The third row is 21∼30 steps. Similarly, in the right box, the shotsnaps from first row to the third row is 270∼280, 280∼290 and 290∼300.

It is illustrated that the firing is highly random and sparse in the initial phase, but is self-organized into coherent firing in the convergent phase (spikes related to certain shapes are likely to appear in one or two close snapshots).

(Middle) The patter in the middle (white background, colored patterns) is the spike plot of neurons in SCS (x-axis: time; y-axis: neuron index).

The spikes for each neuron are colored based on the ground truth segmentation. For example, a spike is colored as green if it is fired by the neuron corresponding to the "green pixel" in the ground truth image (on the left of the left box). Therefore, neurons related to different objects are separated (in y-axis) and colored as distinguished colors for more intuitive visualization.

The population activity for each object-related sub-population is shown by the colored shadow behind the spikes.

From the picture, we can observe that starting from zero-firing in the first 10 steps (population activity is zero), the SCS begins to randomly fire spikes in the following 10∼20 steps (the sub-population activities are wide-spread and overlap with each other). In the convergent phase, sub-population activity gets separated in the time axis (more sharp), indicating neuronal coherence and successful grouping. Note that the grouping is soft since the spikes are allowed to fire at a narrow temporal interval instead of an absolute pre-defined time point (like the slot model).

(Bottom) The black curve at the bottom is the population activity of all the neurons in the SCS, denoted as LFP (short for Local Field Potential) in the main text. In the convergent phase, the LFP acts as emergent temporal reference to coordinate each spike.

**Additional results**

The top plot in Fig.6 shows the SynScore and AMI Score that is averaged over the 200 test samples. Similar to the Fig.5, the score converge fast during initial 60 steps and get further improved after the $T_{train} = 100$ steps.

The bottom in Fig.6 visualize all the coloring results of another randomly selected input in the test set.

The Fig.7 shows the entire shotsnaps over the whole simulation given a randomly selected image.

1 M.

### A.8.2   Details for Fig5(a)(b)(c)

The Fig.5(a)(b)(c) in the main text (copied in Fig.8 and Fig.9 in SI) show the tendency of training Loss, reconstruction loss (of DAE) and Scores during the training epochs. All quantities are evaluated at the end of each training epochs. There are 300 epochs in total.

The Scores in Fig.8 are averaged over 1000 testing samples, each computed based on the last interval (of length 10) in the simulation ($T_{test}$, Table.2). The salt&pepper noise is removed for testing.

In Fig.9 (left), the whole simulation length is 100 for training and 240 for testing (See Table.2). In Fig.9 (left), the testing loss is much lower than the training loss, it may be caused by two reasons. one is that the $T_{test} > T_{train}$. Another is that the salt&pepper noise is removed during testing. The Fig.9 (right) is discussed in detail in the following section.

The regression of scores and loss is illustrated in Fig.10 and Fig.11, which more clearly show the correlation between various evaluation quantities in Fig.5(a)(b)(c) in the main text. For example, the

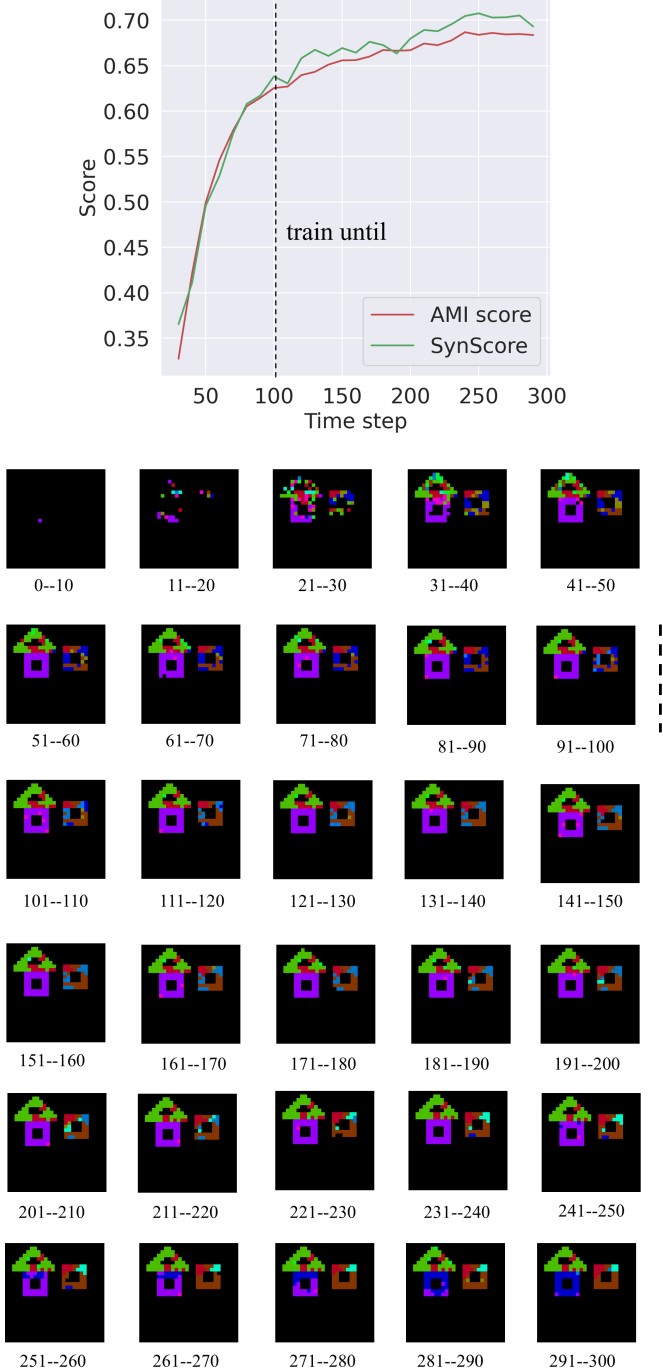

Figure 6: The grouping process of testing images. Upper panel: AMI score and SynScore (averaged over 200 test samples) during the grouping process on testing images (x-axis is time step of length $T_{visual} = 300$). The dashed line indicate the number of time steps used for training ($T_{train} = 100$). It is seen that both score converge fast (achieve 0.6 within 70 time steps) and get even higher after 100 steps (stability of grouping beyond the training length). Lower panel: Coloring of spiking neurons based on the firing times of spikes during the grouping process of an (exemplified) testing image ($T_{visual} = 300$). The number under the image indicate the interval of spike trains used for the coloring. It is obvious that the grouping converge fast in the first 5 intervals and keep stably improving in the later periods. Dashed line has the same meaning as upper panel

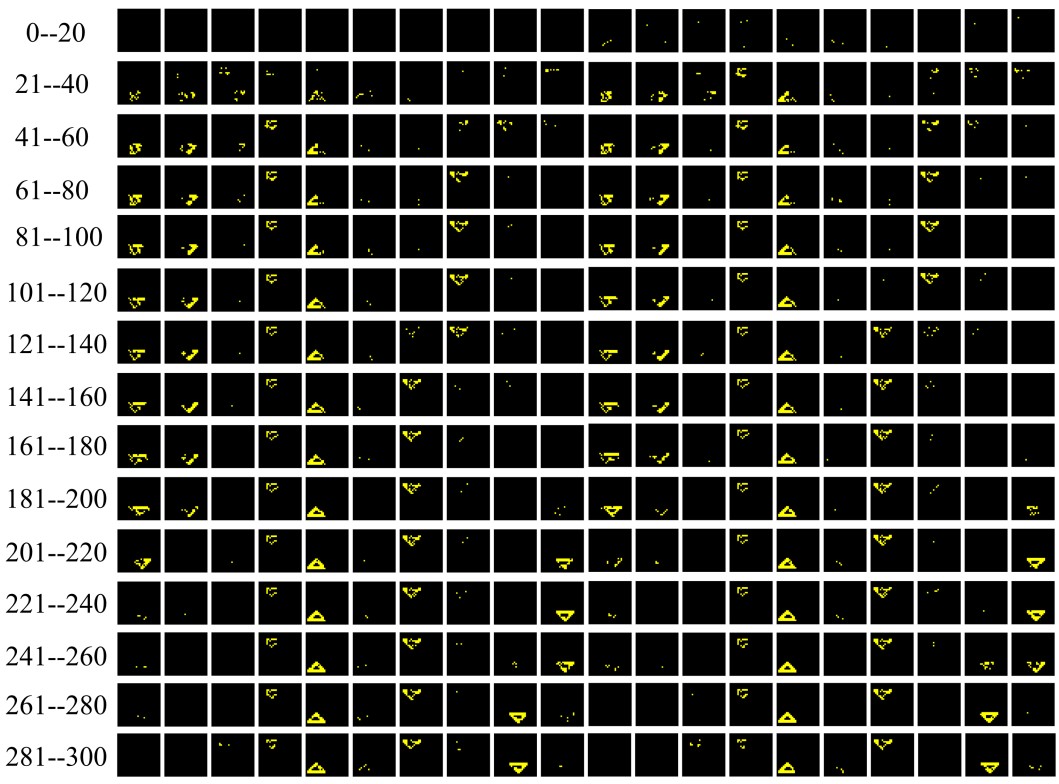

Figure 7: The spiking trajectory along the whole grouping process on a randomly selected testing image ($T_{visual} = 300$, Table.2). Each image stand for the spike pattern at a time step. The patterns are arranged from left to right and from up to bottom. It is clearly shown that objects are separated at timescale $\tau_1 = 2$ and each scene is represented at timescale $\tau_2 = 10$.

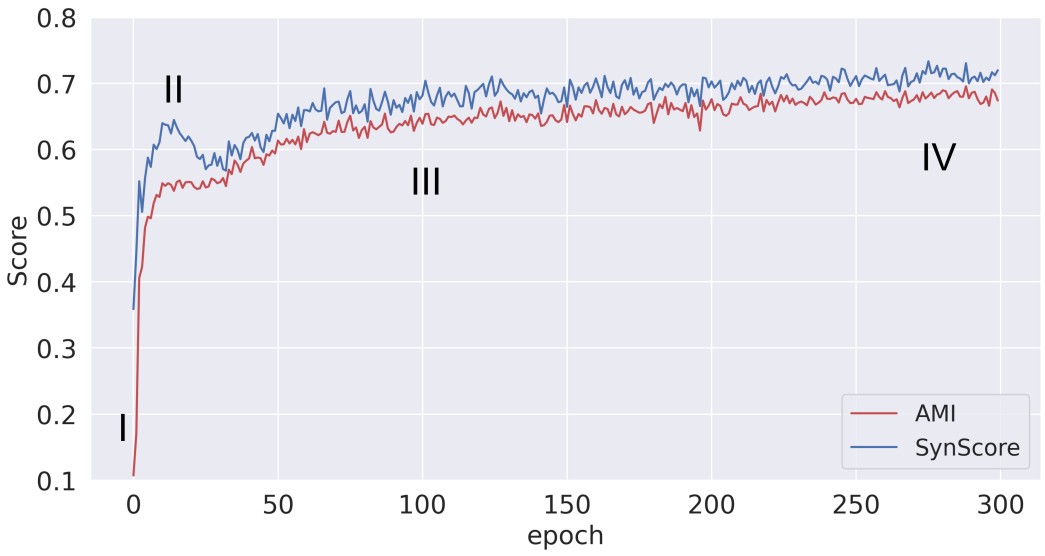

Figure 8: Copied from Fig.5(a) in the main text.

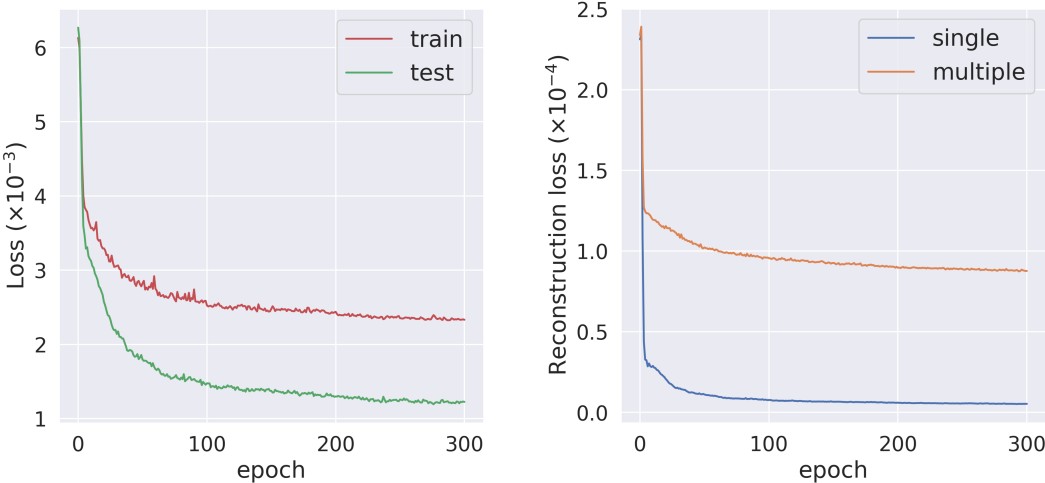

Figure 9: Copied from Fig.5 (b)(c) in the main text.

negative correlation between the scores and the training loss indicates that the grouping ability of spike timing is related to the reconstruction ability of the firing rate (accumulated output of DAE) (Fig.10). The positive correlation between the reconstructive loss of DAE and the training loss suggests that, along the training process, the reconstruction ability of DAE gets improved and the difference between reconstructing single/multiple objects get larger (Fig.11).

**Reconstruction loss of DAE**. The reconstruction loss of DAE (Fig.9right, copied from the main text) is evaluated as the MSE error between the input ($x$) and the output of DAE ($\gamma$) by a single feed-forward pass ($g(f(\tilde{x}))$). The loss is averaged over 20000 samples. Examples of x and outputs of DAE is visualized in Fig.12. The reconstruction losses in Fig.9 (right) illustrate what the DAE learns during the training process. Fig.9 (right) and Fig.11 shows that the reconstruction loss of single objects (blue line) is ten-times lower than the multiple objects (orange line). Thus, the DAE mainly learns to reconstruct single objects (also able to reconstruct multiple objects but with much lower precision Fig.12 right). Since the training loss is only about reconstructing multiple objects, it is not obvious why / how the DAE learns to reconstruct single objects.

Fig.12 shows more details of how DAE learns during the training process: from global features (position, Fig.12 II) to local features (shape or texture, Fig.12 III). It is interesting to note that deep neural network often prefer the latter while the GUST firstly learn the former. In the first phase, the feedback act as "spotlight" attention on a "master map" of features in the feature integration theory (FIT)[23, 44, 45, 46, 47]. It is also observed that during the transition of the two phases, there is a transient period where AMI decreases (Fig.5(a) in the main text or Fig.8 in SI) because compared to a rough but extensive feedback, imprecise texture-level reconstruction may cause confusion due to more localized feedbacks. Fortunately, this intermediate stage only last for a short period (1∼2 epochs). Then, reconstruction improves and AMI increases to a even higher value.

### A.8.3   Details for Fig5 (d)

Fig.5(d) in the main text visualize the SCS representation ($s$) and top-down feedback ($\gamma'$) during the training epochs (same experimental setting as Fig.5(a) in the main text (copied as Fig.8 in SI). One random test example is selected). The last simulation interval of length 10 is considered for the coloring results and each shotsnap (of either $s$ or $\gamma'$).

As explained in Section.A.3, the gating effects of the input refine the firing patterns of $s$ during the phase II, when the top-down feedback is still vague.

Therefore, the object related spikes are more available in a limited observation window (1∼2 steps) so that reconstructing precise single objects becomes probable in phase III and IV.

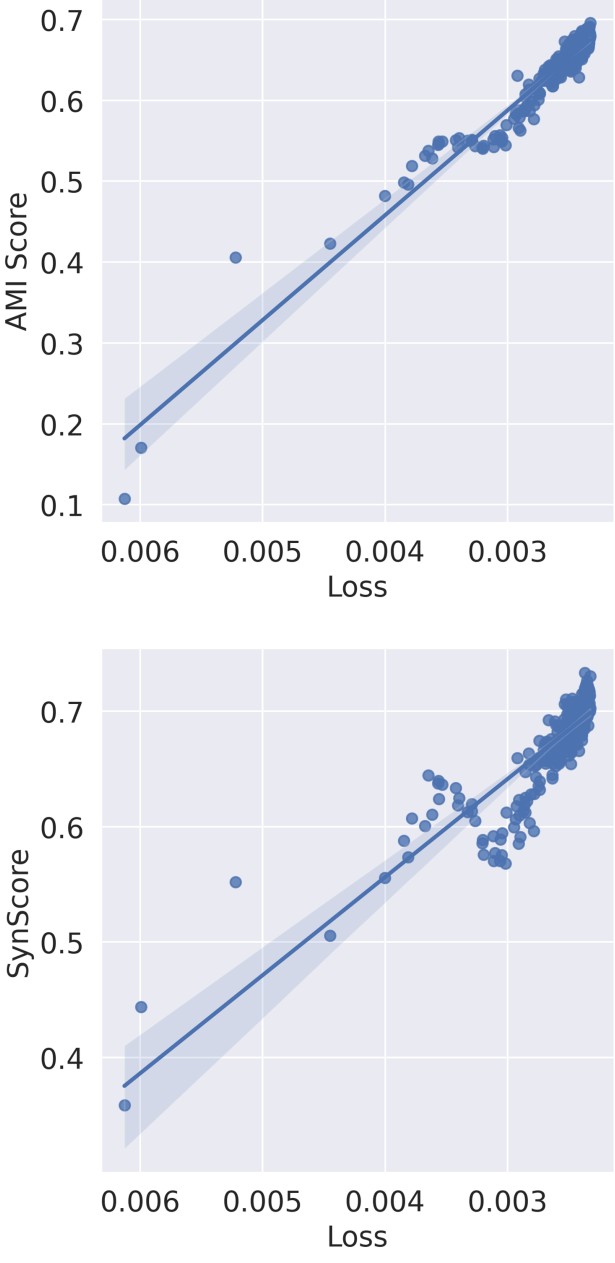

Figure 10: The linear regression of AMI / SynScore and the training loss (inversed order) along different epochs during the training. The negative correlation between the score and the loss imply that the temporal structure emerges and GUST learns to group objects based on the emerged temporal coordinate along the training process.

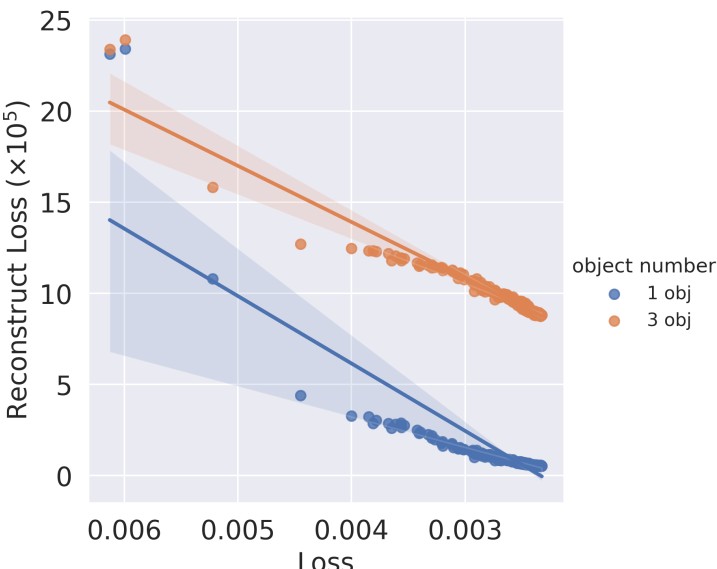

Figure 11: The linear regression of the reconstruction loss (single / multiple objects) and the training loss along different epochs during the training. The positive correlation between reconstruction loss and training loss implies that the DAE learns to reconstruct objects during the training. Note that the minimal reconstruction loss has a ten-fold difference between single-object image and multiple-object image.

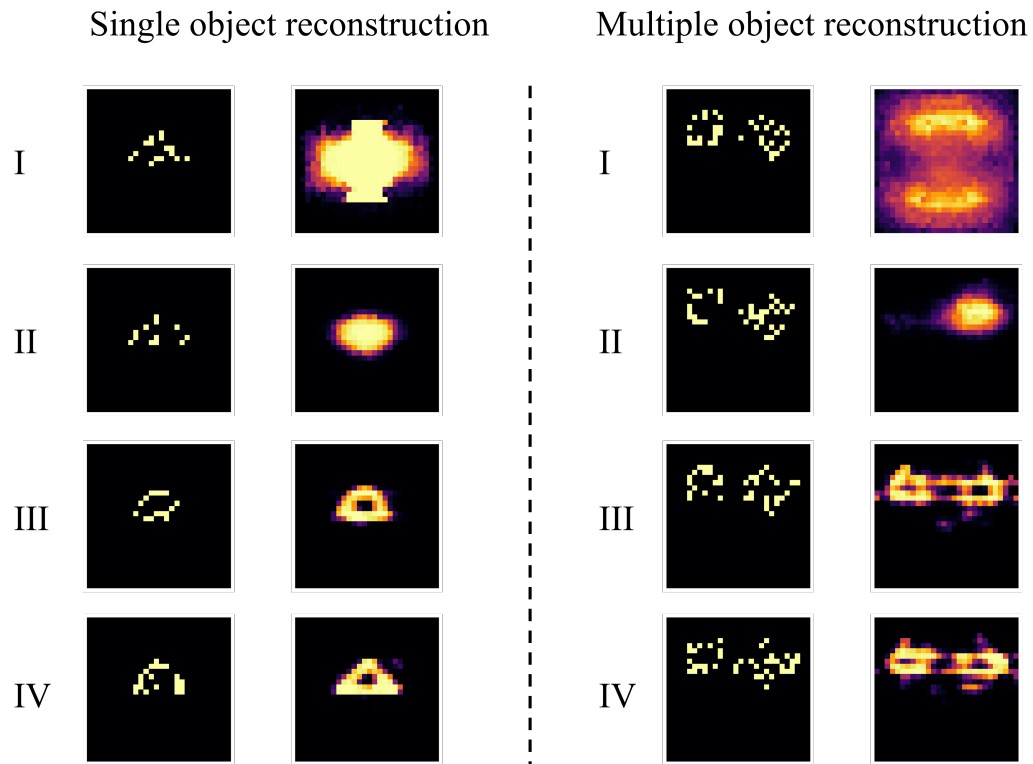

Figure 12: Examples of DAE reconstruction. Left: noisy single-object input and denoising reconstruction of DAE. Right: noisy multiple-object input and denoising reconstruction. From the top to bottom: examples in epoch 0 / 10 / 100 / 300, marked as I / II / III / IV in Fig.8.

Table 3: AMI score and error bars for evaluating combinatorial generalization of GUST. Column is different test object number and row is different train object number

| | GUST | | |
|---|---|---|---|
| # obj. | 2 | 3 | 4 |
| 2 | $0.70\pm 0.0002$ | $0.72 \pm 0.0001$ | $0.73 \pm 0.0002$ |
| 3 | $0.64 \pm 0.0003$ | $0.68 \pm 0.0000$ | $0.73 \pm 0.0001$ |
| 4 | $0.50 \pm 0.0002$ | $0.57 \pm 0.0001$ | $0.59 \pm 0.0001$ |

Table 4: AMI score and error bars for evaluating combinatorial generalization of RC-bind. Column is different test object number and row is different train object number

| | RC-bind | | |
|---|---|---|---|
| # obj. | 2 | 3 | 4 |
| 2 | $0.33\pm 0.0001$ | $0.34 \pm 0.0001$ | $0.34 \pm 0.0001$ |
| 3 | $0.32 \pm 0.0000$ | $0.30 \pm 0.0000$ | $0.31 \pm 0.0001$ |
| 4 | $0.34 \pm 0.0001$ | $0.34 \pm 0.0001$ | $0.34 \pm 0.0001$ |

### A.8.4    Details for Fig.6(a)

In this section, we provide detailed information of experimental setting related to each bar in Fig 6(a) of the main context.

**-noise**. We remove the noise $\epsilon$ in eq .3. More formally,

$$s_i(t) = H(\rho_i(t); \delta_i(t)). \tag{9}$$

**-rfr**. The refractory variable $\delta_i$ in eq.5 has been removed. In other words, $\delta = 0$ for all spiking neurons.

**-delay**. The delay of the modulatory feedback from the DAE output is set to 1 which means $d = 1$ in eq.1.

**-CD**. The coincidence detector is removed which means the time window of it is set to 1. In other words, $s'(t) = s(t)$ in eq.6.

**-spike**. The stochastic neuron in eq.3 is removed. More formally, eq.3 is replaced by:

$$s_i(t) = \rho_i(t). \tag{10}$$

Notably, while the computational graph of the network removes the stochastic neuron (during training), the AMI is still calculated according to the "imaginary" binary firing results calculated by $s_i(t) = H(\rho_i(t) + \epsilon; \delta_i(t))$ for consistency.

### A.8.5    Details for Fig.6 (b)(c)(d)

To evaluate the combinatorial generalization of the representation in GUST, we vary the number of objects in the training set and the test set, ranging from 2 to 4. Here, the combinatorial generalization is defined as the ability of representing scenes composed of different number of objects without ambiguity (the binding problem). This requires the GUST to have the ability of retaining and exploiting the representation of learned single object (entities) to understand the scene of combinatorial structure (number of objects) not seen in the training process.

We generate Shapes datasets composed of 2/3/4 objects, 20000 for training process (18000 for training and 2000 for validation) and extra 1000 random samples for evaluating the AMI scores. Background is considered for evaluating AMI (K=#obj+1). Five random seeds are used and the error bar are very low (around 0.0001, Table.3). Same for the benchmark models including RC-bind (Table.4) and SlotAttention (Table.5). We keep the same network structure and same set of hyper-parameters (Table.2) in all cases. More detailed description of benchmark models is shown in Section A.8.7.

The result (Table.3) shows that AMI scores across different testing dataset are comparable, even if the testing object number is different from the training object number. Besides, all evaluations are based on the same network structure of GUST. It is shown that the more object number used in

Table 5: AMI score and error bars for evaluating combinatorial generalization of SlotAttention. Column is different test object number and row is different train object number

| # obj. | SlotAttention | | |
|---|---|---|---|
| | 2 | 3 | 4 |
| 2 | $0.03 \pm 0.00000$ | $0.06 \pm 0.00002$ | $0.08 \pm 0.00000$ |
| 3 | $0.05 \pm 0.00001$ | $0.05 \pm 0.00001$ | $0.08 \pm 0.00000$ |
| 4 | $0.27 \pm 0.00003$ | $0.31 \pm 0.00002$ | $0.32 \pm 0.00002$ |

training process, the less AMI score is achieved, partially because more objects introduce challenges for learning to segregate raw inputs. Once the segregation is learned, the learned representation of entities can generalize well to different testing object numbers.

### A.8.6 Counter-intuitive common fate of all models

In Figure 6 (b-d) in the main text, it seems like all models, including GUST, slightly perform better for larger number of test images, which seems counter intuitive. One explanation is that object number could have slight effect on overall overlap ratio among objects (Fig.4). Heavier overlap tends to cause more unbalanced cluster size. Since AMI favors unbalanced situation, it tends to have slightly better score. Despite of slight bias, it is still clear that GUST generalizes in different cases ($\sim$0.6) and outperforms benchmarks in each case.

### A.8.7 Benchmark

**RC-bind**.

For comparison, the benchmark model RC-bind[18] is evaluated on the same tasks. RC-bind exploits the feedback of a DAE (single-layer MLP as encoder and decoder, Table.6) that is "pre-trained" to denoise the single objects to further cluster the superimposed objects. The essential difference of RC-bind from GUST can be seen from three aspects. First, RC-bind needs a pre-training process for DAE separately and then combine DAE with a EM-like algorothm[48] for clustering. Thus the grouping ability on the test dataset is not "generalized" from the training because training and testing has different targets (one for DAE only and one for DAE + EM-like algorithm). In contrast, GUST learns to segregate directly in the segregating process (no pre-training is needed). Second, RC-bind requires a hyperparameter K (determined by human) to explicitly separate its latent space for representing different single objects. On the contrary, GUST do not need such prior information and can implicitly group objects with self-organized temporal coordinates. Third, the DAE in RC-bind is pre-trained mainly for denoising single objects. When pre-trained on multiple objects, it is reported that the grouping ability is much worse. We inherit the same network architecture and hyperparameter as in [18] and also use hard-assignment for grouping (Table.6). The only difference is that the hyper-parameter K is set to be the number of objects + 1 (for background) to keep the same as GUST (in original paper, K equals to the number of objects). The size of training / validation / testing dataset is the same as GUST.

**Slot-attention**.

The Slot-attention model[15] also exploits an autoencoder architecture to learn to group objects in an iterative manner. Different from GUST, it uses human-designed discrete slots to "hard" group each object in the latent space. However, the original Slot Attention model is tested only on RGB images instead of binary images. Compared to the RGB images, which may help break the symmetry by the different RGB value of objects, binary images may introduce additional challenges to the Slot Attention model to group the objects.

To implement the slot-attention model, we adopt the open source code of the paper [15]. Some parameters have been changed to fit our tasks. First, because the original input image size of slot-attention is $128\times128$ and our input image size is $28\times28$, the encoder and decoder network sizes have been reduced. We change the encoder network structure to a two layer convolutional neural network and its channel numbers are both 64. The decoder network's channel numbers are also changed to be both 64. Second, the slot number for each tasks has been set to the number of objects (in the train dataset) plus one. Third, the input channel is set to one to deal with the binary input images. More details are shown in Table 7. To calculate the AMI score of slot grouping, we keep the same process

Table 6: Hyperparameters for RC-bind.

| hyper-parameters | configuration |
| --- | --- |
| learning rate | 0.49402 |
| encoder | [l]FC(784, 512) |
| ReLU() | |
| FC(512, 400) | |
| Sigmoid() | |
| decoder | [l]FC(400, 512) |
| ReLU() | |
| FC(512, 784) | |
| Sigmoid() | |
| salt&pepper | 0.9 |
| K | #obj +1(bg) |
| grouping | hard assignment |
| initialization | Gaussian |

Table 7: Hyperparameters for slot attention.

| hyper-parameters | configuration |
| --- | --- |
| learning rate | 0.0008 |
| encoder | [l]Conv(1, 64) |
| LeakyReLU() | |
| Conv(64, 64) | |
| LeakyReLU() | |
| decoder | [l]ConvTranspose2d(1, 64) |
| LeakyReLU() | |
| ConvTranspose2d(64, 64) | |
| LeakyReLU() | |
| slot number | #obj +1(bg) |
| slot size | 64 |
| epoch number | 200 |

as in [15]. More specifically, we assign the category of each pixel according to the largest alpha-mask dimension $d$. $d$ indicates which slot the pixel belongs to. After assigning a category to each pixel, AMI is calculated according to the ground truth segmentation of the image (background is included).

### A.9 Soft grouping

In the main text, we mentioned that compared to a human-designed discrete hard slot in ANN models[17, 18, 14], the grouping with neuronal coherence is soft and the softness helps to encode the uncertainty of grouping.

In Fig.13, an ambiguous input with highly overlapped objects is given to the GUST. As expected, the grouping should be less confident than a non-overlap image. Such uncertainty is reflected in the neuronal coherence, since the SynScore and AMI is around 0.5 in the convergent phase. Besides, Although the objects can still be separated based on spike timing patterns (indicated by the AMI score, coloring results and spike pattern in the black box), the spike firing patterns spread into a "larger" temporal region even in the convergent phase. And the temporal structure (LFP) is less salient. Therefore, the neuronal coherence can be regarded as a natural measure of the grouping confidence and the level of uncertainty. Such softness can be naturally measured by the dendrites of downstream neurons that are sensitive to the relative timing of input spikes[31].

### A.10 Multi-solution

In the main text, we mentioned that: "Firstly, grouping from raw sensory input is in principle an unsupervised task with multiple possible solutions. For example, how to segment a scene or how to

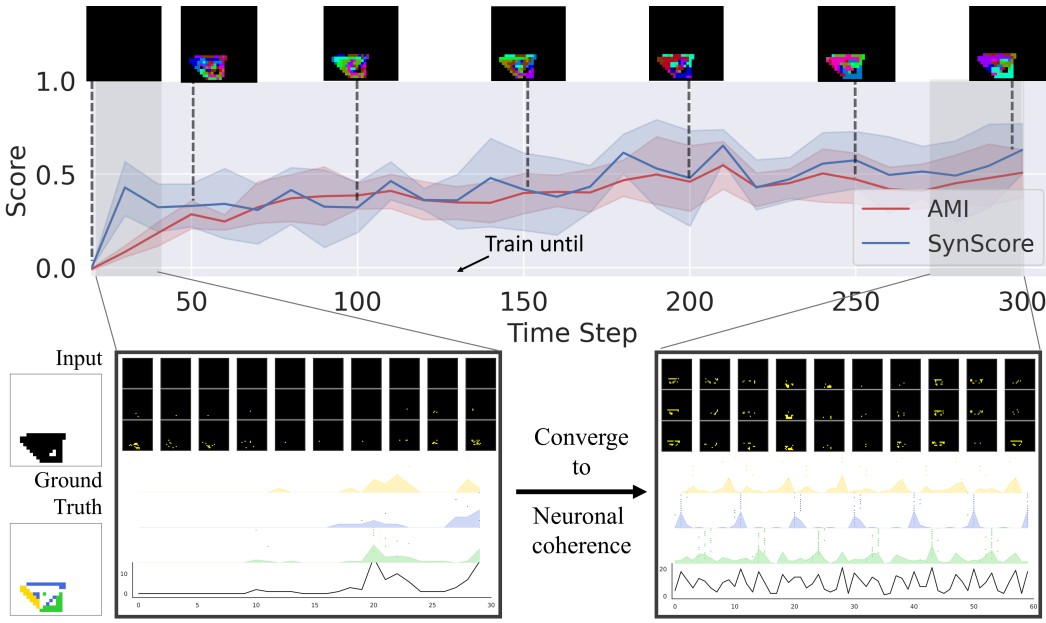

Figure 13: The simulation of grouping with neuronal coherence (similar to the Fig.4 in the main text). An ambiguous images is used.

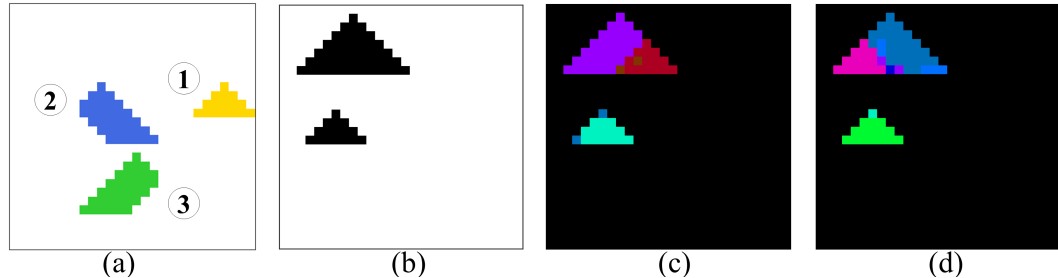

Figure 14: Multi-solution property of GUST. Varified Shapes dataset for better illustration. (a)3 types of new objects(denoted as 1,2,3). (b) The input (c) solution 1 (d) solution 2. The coloring is based on spike timing.

chunk a sensory stream into episodic memories do not have a golden criterion. This excludes the solution through massive labeling and supervised learning."

The dynamical nature of GUST, in principle support the multi-stability, which distinguishes itself from feedforward networks. In this section, we show simulation results to illustrate the multi-solution property of GUST experimentally.

To illustrate more clearly, we take the inspiration from an old Chinese game: tangram. We train the GUST on a similar but slightly varified Shapes dataset, which is also composed three shapes (Fig.14a). Then we test the trained GUST on a very ambiguous input (Fig.14b), similar to the game tangram. Notably, the GUST provides two different but reasonable grouping solution of the input for two simulations (Fig.14c,d).

## A.11 More complicated cases

In this section, we provide the exemplified visualization results of grouping in various more complicated cases (color objects, unseen objects, varying size objects, MNIST objects, moving Shapes) to provide the possibility to generalizing the GUST into more complex situations. The dataset and model structure are kept as consistent with the main text as possible.

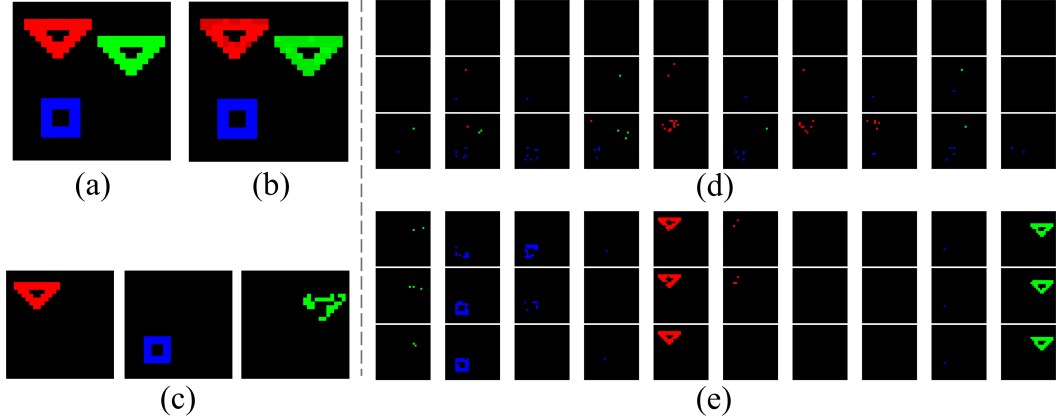

Figure 15: Grouping colored images. (a) input image. (b) coloring results based on spike timing. (c) decoding results based on the clustering of SCS neurons. Three colored objects are decoded. (d) spike recording in the initial phase. (e) spike recording in the convergent phase. In (d)(e), neurons (or spikes fired by each neuron) are colored based on the color range they represent.

### A.11.1   Color input

As explained in Section A.1, the binary input in the main text could potentially be generalized to colored images, though brain-inspired coding strategy like CIE[4, 5]. The resolution of the color encoding can be flexibly chosen. For example, during the evolution, animals have adapted a sensitivity map over the color space (CIE[4]), so that some color / wavelength range are represented with more neurons than others. In this way, the problem of dealing with real valued color input is transformed to scaling the spike coding space (SCS) in an additional dimension to account for discretized zones in the color space. The remaining question is: should the stacking of spiking neurons in the additional dimension harm the learning and dynamics of the GUST? If the GUST can still work, then there is no fundamental issues preventing applying such strategy for dealing with color images in the future.

To this aim, we train / test the GUST on a modified Shapes dataset, where each shape has a color. We discretize the color space into three zones: Red, green and blue[2]. For simplicity, the object are also colored as the three representative colors (Fig.15a). As shown in Fig.15, grouping of color image is consistent with the result in the main text, indicating that the coding strategy[4] is a plausible solution to generalize binary input to more real-world colored inputs.

### A.11.2   Generalize to unseen objects

In this section, we show that GUST can not only generalize with respect to object number, but also generalize to unseen objects. As shown in Fig.16, we test the trained GUST (the same one in the main text) on an image containing two novel objects: a heart and a rectangular (Fig.16a). Surprisingly, the GUST can still group the objects coherently (Fig.16c,d,e) in a zero-shot manner.

### A.11.3   Different size

We also study how variable factors like size influence the grouping, in terms of training and testing. In Fig.17a, we train and test GUST both in a varied Shapes dataset, where the Shapes are randomly scaled with different ratio, max/min=1.5 (too big size may cause severe overlap). As a result, the varying factor in training and testing dataset is more complex: shape type, location, size. As shown in Fig.17a, the grouping result remains intact, indicating that the learning of GUST does not degrade in front of multi-size Shapes.

We further train and test the GUST with different size range to show the effect of size for the testing process: The size in training dataset: max/min=1.5 and in testing dataset: max/min=2. Therefore the

---

[2]We first change the input size of the network to $C \times H \times W$ where $H$ is the width of the input image, $W$ is the height of the input image, and $C = 3$ means the number of the possible discrete colors. Meantime, the hidden layer is modified to 1200 because of the larger input image size. The other parameters remain unchanged.

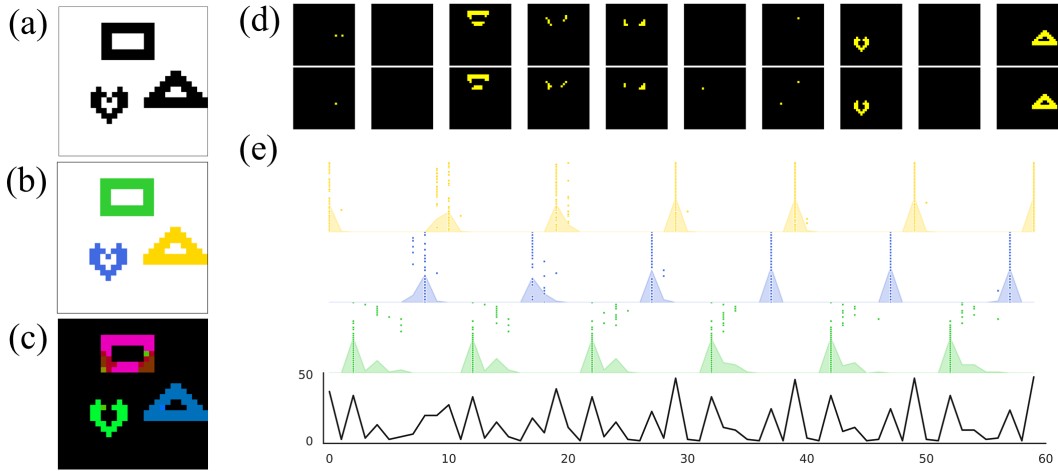

Figure 16: GUST generalized its grouping ability to unseen objects. (a) input image. (b) ground truth of input image. (c) coloring results based on spike timing. (c) spiking pattern. (d) spike recording. (c)(d)(e) are based on convergent phase. Note that the heart and rectangular are not seen in the training data.

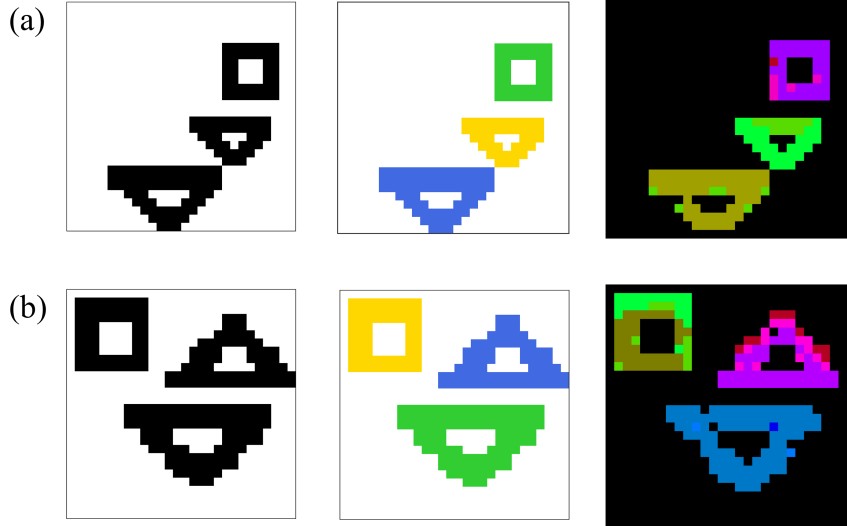

Figure 17: (a) The grouping result of GUST trained and tested on a varied Shapes dataset, where shapes are scaled in different ratio (max/min=1.5). (b) The GUST generalizes to bigger Shapes (max/min=1.5 in training, max/min=2 in testing). From left to right: input, ground truth, coloring results based on spike firing time.

Shapes in the testing set is generally larger than the training set. As the result, the GUST needs to generalize its knowledge about size (acquired in training) to deal with bigger shapes in the testing images. The results in Fig.17b shows that the grouping of GUST generalize with bigger Shapes.

### A.11.4 Grouping multiple MNIST

We also shows the results on multi-MNIST dataset (both train and test), which is composed of much more complex objects. The successful grouping results are shown in Fig.18. The only slight change of GUST is that: the hidden layer size is modified to 700 for this more complex dataset.

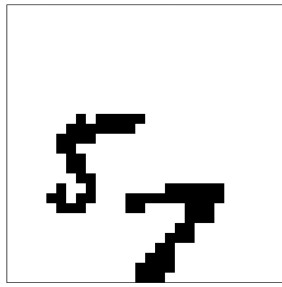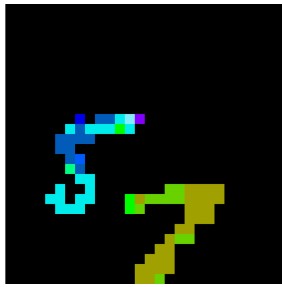

Figure 18: The grouping results on MNIST dataset (a) input image. (b) coloring result based on spike timing.

### A.11.5 Grouping moving Shapes

Temporal grouping inherently implies a contradiction: the time is used for dual roles. On the one hand, it is used to represent a single state of the scene, eg. to separate objects. On the other hand, if object can move, time needs to describe its dynamics or trajectory. Since the grouping takes time, if the object moves continuously, the representation is not able to track it continuously. To this end, we additionally show whether the GUST is able to group moving objects in the temporal dimension. Will the dual use of time degrades the grouping?

We assume a separable time scale $\tau_1 \gg \tau_2$(Fig.1 in the main text), and the object only moves at the slower timescale $\tau_2$. This assumption is consistent with the human perception. If we take the $\tau_1$ as the timescale of temporal coding, eg. synchrony, which is about 10 ms[49], the finest movement we can perceive has timescale more than 100ms[50]. This temporal scale separability provides the essential basis for temporal grouping in general situations.

The motion of object is formulated as a stochastic process. At each time point, the object has a probability $p_{go} \in (0, 1)$to move in a random direction and that of $(1 - p_{go})$ to stay. By controlling $p_{go}$, we can determine the expected distance that object moves during each time step. Effectively, the object moves at an arbitrary continuous slow speed in the grid world.

For the moving Shapes case, we inherit the same network architecture and parameter as static case in the main text for model consistency. The GUST is trained with the same training/testing protocol as before except that the averaged firing rate is compared to the averaged input traces (during last $\tau_2$ steps).

$$L(x(t)) = (H(\sum_{t=T-\tau_2}^{T} x(t)) - \frac{1}{Z} \sum_{t=T-\tau_2}^{T} \gamma_i'(t))^2 \tag{11}$$

H is the Heavi-step function of threshold 1, so that all traces the Shapes pass during $T - \tau_2 \sim T$ is set to be 1. $\gamma_i'(t)$ is computed based on noisy input $\tilde{x}$. Z is a normalization factor as in main text. The practical procedure we use for normalization is that both $H(\sum_{t=T-\tau_2}^{T} x(t))$ and $\sum_{t=T-\tau_2}^{T} \gamma_i'(t)$ are normalized to be a distribution (of size $28 \times 28$, sum=1). In this way, two quantities can be compared by MSE. Z is just an effective description for this process. In sum, the loss function is a minimal extension of that in main text.

The visualization of the spiking pattern of feature neurons (during discrete time intervals of timescale $\tau_2$) and averaged inputs (during the same time interval) is shown in Fig.19. The coloring results shown that, the slow-moving objects are still being grouped. However, the continuous motion (Fig.19 bottom) is represented discretely at timescale $\tau_2$ (Fig.19 up), by referencing to self-organized discrete temporal coordinate (Fig.1 in the main text).

As more clearly shown in Fig.20, While the x(t) in the lower panel "continuously" move in time, the representation of S(t) is discrete. For example, a complete representation of a single scene may take $\tau_2$ time steps (half a row in the upper pannel of Fig.20), because different objects are represented at different time points. However, during this temporal interval, the objects have a probability to change

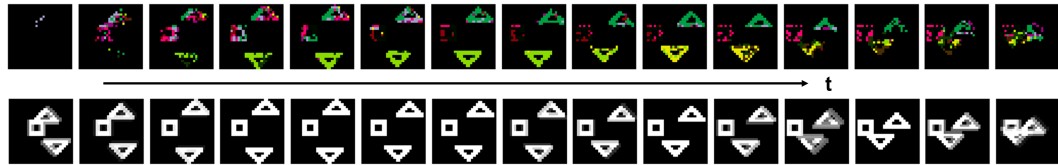

Figure 19: Visualization of temporal grouping of moving Shapes. Upper: coloring results of the spiking trajectories based on firing times; lower: averaged inputs. Input objects are colored as white and the background black, for clear visualization.

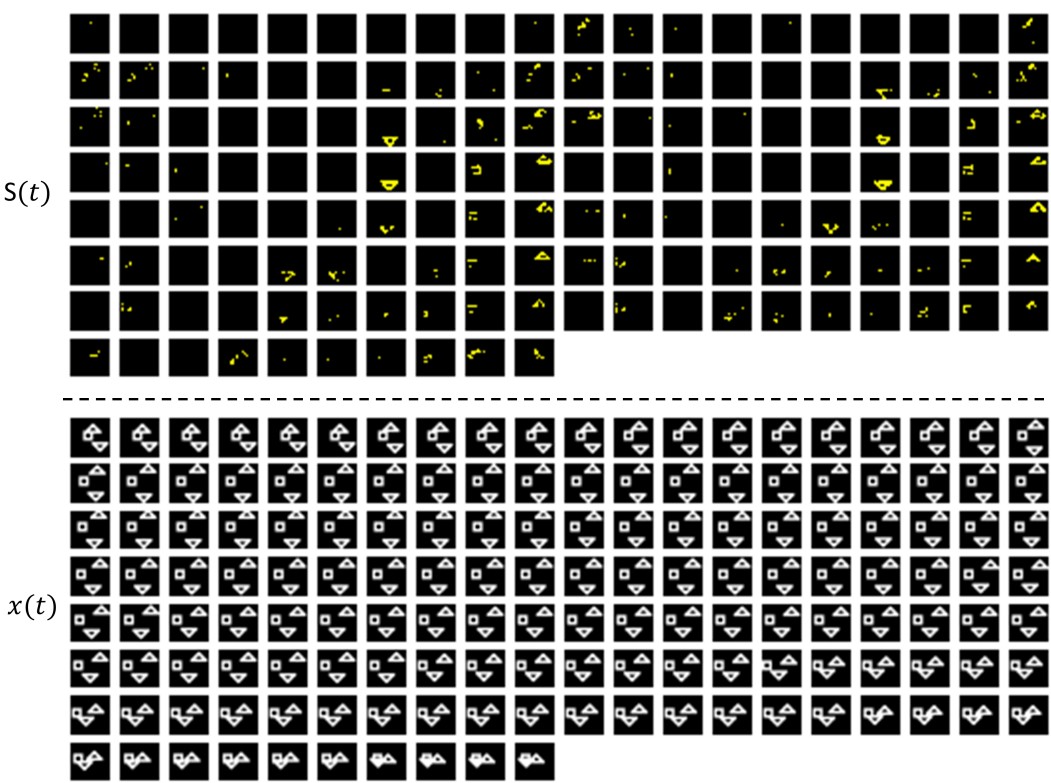

Figure 20: Spiking pattern S(t) (upper) and input x(t) (lower) during the whole simulation of length 150 time steps. Each image stand for a single time step. images are arranged from left to right, from up to bottom.

their states (move). If this occur, the representation will lag behind the input x(t). Thus, the grouping representations are discrete "snapshots" of the moving object at timescale $\tau_2$.

Interestingly, the result implies that there is an inherent constraint of the temporal resolution of perception for temporal grouping models. The perceptual constraint is not due to the sampling limit of receptors, eg. Nyquist sampling theorem applying to retina[51], but a result from the limitation at the representational level itself. Since it seems that the brain and the GUST share a common limitation on temporal precision[50], it is interesting to discover whether biological system also operate in a discrete manner as GUST in the future work[52].

## A.12 Evaluation and metric

In this section, we introduce the metric and evaluation method in detail.

$$D_{VP}(s_i, s_j) = \min_T \left( \sum_{u=1}^{|T|} cost_{T(u)} \right)$$

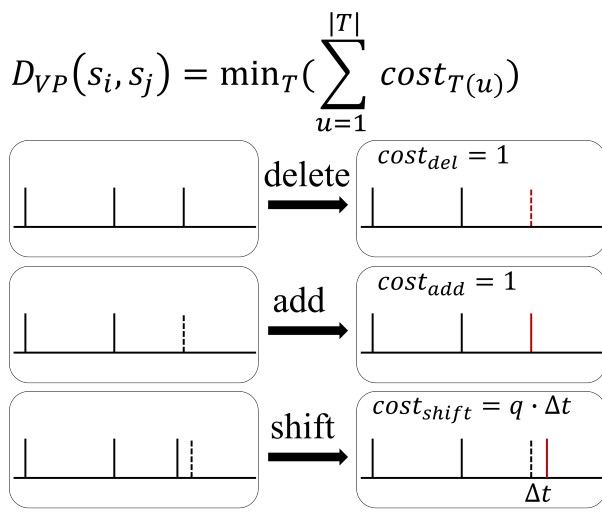

Figure 21: Illustration of the Victor-Purpura metric, copied from main text Fig.5. $T(u) \in$ {add, delete, shift}

### A.12.1 Victor-Purpura Metric

The Victor-Purpura metric is a classical non-Euclidean metric to measure the distance between arbitrary spike trains for evaluating the temporal coding in the visual cortex[49]. Three types of operations are identified (Fig.21): 1. add a spike (cost=1); 2. delete a spike (cost=1); 3. shift a spike for length $\Delta t$ (cost=$q \cdot \Delta t$). By sequentially applying the three operations, an spike train can be transformed to the other. $q$ is a parameter to control the temporal precision. The Victor-Purpura distance is defined as the minimal cost to transform a spike train to the other. It is proved that the definition satisfy the three principles of a metric: positivity,symmetry, and triangle inequality[49]. Thus, it induces a metric space of arbitary spike trains, even not embedded in a vector space of finite dimension.

If the q is chosen to be 0, then shifting a spike will cause no cost. Thus the distance is exclusively due to spike count, therefore spiking rate is measured. If the q is chosen to be large, eg. infinity, then it measure the number of spikes that is not in absolute synchrony (total synchronous spike trains have 0 distance while slight shift of spikes has cost 1). Thus, $\frac{1}{q}$ is treated as a timescale parameter, to control the precision of temporal coding. In this paper, q is cosen to be $\frac{1}{3}$, indicating a timescale of 3 time steps.

### A.12.2 K-Medoids clustering

To evaluate the grouping quality, the spiking pattern needs to be "readout" by a clustering method. In other word, the GUST grouping object into implicit spike timings and a clustering process make the grouping explicit and visible. Since the grouping assignment is given by precise spike synchrony, the metric for measuring the distance of spike trains should reflect precise spike timing information. To this end, Victor-Purpura metric is applied (Fig.21). The clustering process treats each neuron i as a sample and the spike train $(s_i(t))$ as the feature of the sample. By clustering neurons based on VP-metric of their spike trains, synchrony is readout and the cluster is interpreted as groupings.

Since the ground truth is available for Shapes dataset and the number of objects is known in advance, it is convenient to use K-means for clustering. However, K-means is constraint in Euclidean space where the mean is iteratively updated as cluster centers, which is not compatible with Victor-Purpura metric. For example, we can not measure the distance between a spike train and an averaged mean by VP-metric. Thus, we prefer a clustering method which is defined in sample space only. A direct alternative is the K-Medoids method[53], which finds optimal cluster centers within the samples in each clustering iteration. Therefore, we combine the K-Medoids and the Victor-Purpura metric to readout the implicit grouping information for the following quantitative evaluation. More formally (same as Fig.3 in the main text),

$$s^c = \text{argmin}_{s \in I(c)} \left( \sum_{s' \in I(c)} D_{VP}(s, s') \right) \tag{12}$$

$$I(c) = \{s | C(s) = c\} \tag{13}$$

$$C(s) = \text{argmin}_{c \in C} D_{VP}(s^c, s) \tag{14}$$

Here, $C$ is the set of $K$ cluster values. $C(s)$ is the clustering assignment for each spike train s. $I(c)$ is the "inverse" of $C(s)$, which describe the set of spike trains in each cluster $c$. Such clustering assignment is processed iteratively until convergence. In this paper, only the spike train during the "convergent phase"(last 10 steps) is selected for clustering. Longer length may increase the quality of K-Medoids clustering (because spike train is longer), but 10 (identified as $\tau_2$) is enough for clustering.

### A.12.3  AMI

Now, we have readout the explicit clustering result from implicit spiking patterns. To evaluate the grouping quality, the mutual information between the clustering result and the ground truth is measure by adjusted mutual information (AMI)[54]. The AMI is independent of absolute cluster values, thus a permutation of cluster values does not influence the score. Thus, it is suitable for evaluation in this work, because the cluster result might be a permutation of ground truth assignment. Higher AMI score implies more mutual information between the cluster and ground truth. 1 stand for perfect grouping quality and 0 stand for chance level grouping quality.

### A.12.4  SynScore

The neuronal coherence indicate the saliency of the emerged temporal structure (coordinate), which imply the precision or confidence of the grouping. To measure such temporal structure, we identify neuronal synchrony as inner-cluster coherence (of K-medoids clustering) measured by the same Victor-Purpura metric. More specifically, we combine Silhouette score[55] and VP-metric to measure the neuronal coherence. The Silhouette score is one of the methods to evaluate such inner-cluster coherence. The higher score means closer distance between spike trains in the same cluster, and thereby implies that neurons are more synchronous inside each cluster (more salient oscillation and temporal structure).