# OpenReview forum: "GUST: Combinatorial Generalization by Unsupervised Grouping with Neuronal Coherence"
_NeurIPS.cc/2023/Conference — NeurIPS 2023 poster_

### Official Review · Reviewer_9PtM · 2023-07-04

**Soundness:** 3 good
**Presentation:** 3 good
**Contribution:** 3 good
**Rating:** 5
**Confidence:** 4

**Summary:**

This paper uses findings from vision sciences research to provide an image-segmentation model that could be used in ANNs. The “grouping with temporal coherence” assumes that grouping information emerges via spiking synchrony. This is implemented in their network, called GUST. The network is composed of two main parts: a spike-coding space (SCS) and a denoising auto-encoder (DAE). The DAE provides delayed feedback to the SCS. Unsupervised training enables GUST to detect objects with neuronal coherence. The neuronal coherence is evaluated through a metric that measures distance between spike trains of populations of artificial units conceptualized as neurons. The system is iterative and is shown to generalize as well.

**Strengths:**

The paper attempts to address a very important question in computer vision/neurosci. The network is brain-inspired and does not require extensive training. GUST shows through training that it gradually learns to segregate the scene into a self-organized temporal structure and represent the building blocks of a scene by “neuronal” coherence. This can have major implications for computer-vision models and possibly be used to resolve the difficult problem of identifying objects in cluttered scenes.

**Weaknesses:**

Although this work has a broad audience, the framing of the question is rather poor. The authors start by talking about the binding problem, but then they constantly give examples that are about cluttered visual scenes. The binding problem is more general and can be applied to e.g., grouping of points that belong to a circle.
Some assessment of the sensitivity of their model vs. for example RC-bind is missing (e.g., showing the performance as a function of %overlap). The comparison is only made for generalization to a different number of segments (Figure 6). Also, they don’t explain whether/how this can be combined with deep-net models of visual object recognition.
From the neuroscience point of view, the authors seem to take for granted that the neuroscience evidence supporting perceptual grouping or binding by synchronized activity of neurons, but in neuroscience this is far from being accepted. It is indeed controversial. While it is Ok to present it as a neuroscience theory with credit and evidence in favor, it should also be mentioned that this theory is controversial and that there are alternative theories. Better balance would be highly beneficial.

**Questions:**

What exact computer-vision problem is this resolving? It should be clear that this paper is about image segmentation and not perceptual grouping. Perhaps the most relevant literature in neuroscience is object recognition in cluttered scenes.

Why could this not be accomplished by deep neural network models with a lot of training data? A simple intuitive answer is lacking.

Under what circumstance does the grouping by temporal coherence fail in realistic cases? For example, choices like tau2 might not be wise or it might fail in speedy segmentation.

In Figure 6 (b-d), it seems like all models, including GUST, slightly perform better for larger number of test images. This seems counter-intuitive to me. Can the authors explain why the performance does not drop in their model?

I find it very interesting that the “neuronal” coherence could encode the grouping uncertainty. This feature is not exploited in any of the simulations though. Do you think that you can emphasize more this?

**Limitations:**

The limitations are not deeply or explicitly elaborate. This is problematic. Moreover, existing alternative algorithms are mentioned, but they are not systematically compared against the new work presented by the authors.

---

> ### Author Rebuttal · Authors · 2023-08-06
>
> __A1. Framing of the question__.
>
> The major problem this paper focuses on is indeed the binding / grouping problem (in ANN / neuroscience literature), which is clearly framed in Section 2 in main text. The review paper [7] in main text frames binding problem as representation, segregation and composition
>  (__Fig 3 in global pdf__) and this paper focuses on the first two, which is central to the binding problem.
>
> It seems that what really concerns the reviewer is the relevance between the grouping problem (‘more general’ as reviewer argues) and the task we demonstrated (‘cluttered visual scenes’/’image segmentation’/'classification' as reviewer suggested). On the one hand, the binding problem _in neuroscience_ is famously exemplified as grouping features of cluttered visual objects, so called superposition catastrophe (See ref [6][7][11] in main text, copied in __Fig 3 in global pdf__); On the other hand, studying binding / grouping problem via cluttered visual scenes is common in a list of works in _ANN field_ on binding (See [14][48][7] in the main text). While the task seemingly bears the similarity with image segmentation / object discovery in ANN field, they focus on different challenges (we design GUST for general grouping problem challenges [unsupervise, multistable, common format...] with desirable features of general sense [denoising mechanism], instead of to conquer specific static image segmentation). Since the architecture in this work do not have explicit constraints / expert knowledge on the modality/semantics of data (eg. classical image segmentation algorithms like MRF or supervised segmentation), the mechanism could be generalized to other modalities or more abstract high-dimensional representational space (argued in Section 2 in main text). Thus, the visual task in this paper is for vividness and should not loss the generality. For a systematic review, please see ref [7] in main text, which could resolve most of the concern about the background knowledge (how to ground binding problem in ANNs field, w.r.t related topics like segmentation and object discovery, in neuroscience, and in Gestalt psychology). For example, as to the reviewer's __concern__: ’ _grouping of points that belong to a circle_ is also binding', it is actually specific principles (proximity, similarity, closure, symmetry, etc) in Gestalt Psychology. However, in Gary’s work [1], it is argued that various Gestalt Laws are all special cases of a single information-theoretic grouping principle, which is what we focus on.
>
> [1] Gary Hatfield and William Epstein. The status of the minimum principle in the theoretical analysis of visual perception. Psychological Bulletin, 1985.
>
>
> __A2. Relation to deep learning__.
>
> __The intuitive answer of why grouping challenges deep-nets__ is explained in the second paragraph of Section2 in main text and reviewed in detail in ref [7] in main text. On the specific question of why large data can not solve grouping is mentioned in the paper (line28): ‘the cost of representing all possible combinations statically and locally is exponentially high’. While the specific task seems easy to be conquered by specific ANN models with supervised training, we focus on solving the task with the model of desirable feature that is of general sense, so that it has potential to solve a family of tasks( see A1), which is a hard problem.
>
> __The way GUSTs could be combined with deep-nets__ is diverse and is explicitly discussed in A.1 of the appendix (line 76, 99). Briefly, generative model and denoising is very basic idea in deep learning, indicating possibility to link to more advanced models (eg. diffusion model). Also, the spiking activity in SCS could be readout to ANNs (to solve downstream tasks like _recognition_) by HU proposed in ref [56] in main text.
>
> __A3. Concern for more results__
>
> For analysis of __overlap effect, see A.3 in global response__. Besides, as far as we understand, for single level grouping, varying 'object number' is complete for composition. For more grouping results, see A.12 in appendix.
>
> __A4. Controversial of binding by synchrony__.
>
> Actually, two main-streams of binding theories are synchrony-based and attention-based (FIT theory), which is briefly mentioned in appendix (line123). We actually combine the two sides in our model. We will balance the two sides better when we revise the paper.
>
> __A5. Failure case__.
>
> Failure case has been shown in Figure 12 in appendix, when overlap is sickly heavy. Moving case is also challenging (dual use of time, See A.12.5, Figure 18 in appendix). In A.12.5, we also discussed the _speedy_ consideration (line 538, line545, line 574). For $\tau_2$ , as Figure 6 in main text shows, same $\tau_2$ accounts for grouping varied number of objects. It is the grouping dynamics that has the 'wisdom' (flexibility).
>
> __A6 Counter-intuitive common fate of all models__:
>
> Object number could have slight effect on overall overlap ratio among objects(__Fig1 in global pdf__). Heavier overlap tends to cause more unbalanced cluster size. Since AMI favors unbalanced situation, it tends to have slightly better score. Despite of slight bias, it is still clear that GUST generalizes in different cases (~0.6) and outperforms benchmarks in each case.
>
> __A7 ‘Grouping uncertainty’__.
>
> The coherence level can be readout by a coincidence detector (HU in ref[56]) to help downstream tasks. In general, such coherence acts as an internal indicator of 'good' solutions so as to enable communication / transmission to downstream processes. For concrete example, see __A.8 in global response__.
>
> __A8. Response to limitation__.
>
> The limitation is explicitly elaborated in A.1 in appendix. We compared our work with RC-bind and slot attention as benchmark in this paper. In A.9.6 in appendix, we provided more detailed discussion of their relevance. While Slot attention is one recent SOTA on color image, we can add more related models when we revise the paper.

---

> > ### Comment · Reviewer_9PtM · 2023-08-11
> >
> > Thanks for these answers. In the light of the replies and of the work done in response to requests of others, I am happy to raise the score to 5. I remained concerned about the apparent lack of awareness of the authors that the binding or grouping by synchrony theory is at best controversial in neuroscience (see e.g. the recent review of Roelfsema in Neuron 2023). The authors should  give a less biased view.

---

> > > ### Author Response · Authors · 2023-08-11
> > > **Response to Reviewer 9PtM**
> > >
> > > We are grateful for the reviewer to introduce the review paper, which provides recent evidences against the temporal binding theory. We are aware that binding by synchrony is controversial in neuroscience all the way along the history and we are happy to take the advice. Specifically, we have provided a more balanced review in the _Related Work Section_  in the revised main text.  Besides, we also aim to integrate synchrony and attention in a single binding framework in future, so as to account for more general binding. Thus, the introduced paper is very helpful for us.

---

> > > > ### Comment · Reviewer_9PtM · 2023-08-21
> > > >
> > > > Thanks to the authors for the continued effort to engage in productive discussions which improve the paper. After having read replies to other reviewers, I have also raised the score of the contribution field from 2 to 3.

---

### Official Review · Reviewer_WxQh · 2023-07-07

**Soundness:** 4 excellent
**Presentation:** 4 excellent
**Contribution:** 4 excellent
**Rating:** 8
**Confidence:** 3

**Summary:**

The paper introduces GUST (Grouping Unsupervisely by Spike Timing network), a network architecture inspired by the human brain that aims to address the challenges of grouping sensory information in artificial neural networks (ANNs). The network incorporates biological constraints to bias the network towards a state of neuronal coherence, reflecting grouping information in the temporal structure of spiking activity. GUST is evaluated on synthetic datasets and demonstrates the ability to learn and represent grouping information from unsupervised objectives. The model progresses from perceiving global features to capturing local features and systematically composes learned building blocks to represent novel scenes. The paper highlights the advantages of grouping with temporal coherence, such as flexibility, dynamism, and reduced representational cost. GUST overcomes challenges related to non-differentiable dynamics, high temporal resolution, unsupervised learning, and explainability. It contributes to bridging the temporal binding theory with ANNs and enables the grouping of objects directly from multi-object inputs. The paper also presents a clustering method to evaluate neuronal coherence based on precise temporal structures.

**Strengths:**

- The paper very clearly presents the formalism in that feels very intuitive and tractable. It is well written and clear. Despite how long the appendix is, the authors did an excellent job of keeping things self contained where limited reference to the appendix is needed to understand the paper

-The paper incorporates really interesting biologically inspired structures into the architecture. Most interestingly, they incorporate the udse of coincidence detectors in a spiking model convolved with feedback information from the DAE to build a model which can group representations of object shapes in an unsupervised way. It is a really interesting architecture and I feel the model is an insightful architecture.

- The background presented is very well motivated. It discusses experimental findings and key results in the field on the experimental and the modeling end which motivate the work in the rest of the paper.




**Weaknesses:**

- While this paper is very technically involved (and I did not read the whole appendix as a 26 page appendix is a bit unreasonable), I did thoroughly enjoy this paper and the ideas it integrates into the model.

- The authors do not appear to compare to some comparable architectures on this task with the same metric. But this is a minor weakness. It is a little difficult to know how to orient oneself when reading this model relative to the literature. However, the novelty of the architecture is more interestng.



**Questions:**

-The modeling of the CD seems a bit confusing to me. The authors call it a coincidence detector but it appears that s' is 1 when a given neuron has rapidly tired between two time steps (this is also confusing to me since the neurons have refactory periods so my understanding is if s_i(t) = 1 then s_i(t-1) can't be 1 . If this is the case, the term coincidence detector seems a bit misleading if it is considering the coincidence of one neuron across time? Or is it the coincidence of multiple neurons? Which would made sense in the neural coherence picture, but if this is the case then should it be s' = CD(s_i(t),s_j(t-1))? Could the authors clarify this? I'm not sure if I am misunderstanding something or if it is a confusion of notation.

-In section 3.2, the authors state: "The refractory dynamics provides the essential temporal competition to separate groups of different objects and acts as a structural bias for a grouping solution". Could the authors better clarify what they mean by this? How does the refractory period provide competition?

- Small typo in header of 3.5. Gredients should be Gradients.

- Could the authors elaborate and clarify why "The GUST is simulated 3-times longer than training to confirm the stability of the grouping.". This is a technical thing I am missing, but if the authors stop training at a particular time, then what is it that continues to change - is it the weights in the DAE are frozen but the model continues to receive inputs to output a score?

- In figure 5d right, where it shows the spike firing sequence, to be very clear, are the individual columns the same neuron firing across time after that particular epoch? Or are they individual neurons (so 10 neurons) firing.



**Limitations:**

The authors thoroughly address limitations in the appendix.

---

> ### Author Rebuttal · Authors · 2023-08-06
>
> We appreciate the reviewer to spend the valuable time for carefully reviewing the paper and providing the inspiring comments and helpful suggestions. We appologize for the confusions the paper raise to the reviewer, which will be clarified as followings.
>
> __A1. The response to concerns on related works__.
>
> In this paper we provide the RC-bind and slot attention model as the benchmark, which is of comparable recurrent architecture (attentional feedback) and of similar metric (AMI). As discussed in the related work (Section 5), the general computational problem is related to object-centric representation or object discovery in CV. Therefore, the general architecture is more or less related to this line of models (‘slot grouping’ in Section 5). RC-bind and slot-attention are two representative models along this line. However, as the reviewer points out, the architecture of this work bears the novelty compared w.r.t this line of works in deep learning, partly because they have different representation assumptions (pre-designed slots vs emergent neuronal synchrony), which is central for binding problem.
>
> __A2. The response to the concern about CD__.
>
> The coincidence detector in neuroscience literature stresses two things: First, the decay is very fast so that only coincident arriving spikes within a very narrow time window could be integrated; Second, the threshold is low so that the detector neuron can be activated by only a few spikes within the integration window. Here we take these ideas: the integration time window is narrow ($\tau_w=2$) and the threshold is low (one spike is able to trigger the neuron). That is partially why we term this non-linearity as CD.
>
> However, as the reviewer concerns, in our model, each single CD (for $s_i$) is applied in such a simplified case that the original interpretation degrades into a short-term integrator of $s_i$ itself, with narrow time window and low threshold. However, all CDs (taken together) still acts as a coincidence detector / coincidence filter of the _population activity_ of SCS. The inputs to the DAE (for non-linear processing), is not a single slice of spiking patterns but narrowly integrated spikes (by CD). So that only coincident spikes $s_i(t), s_j(t-1)$ is taken into account by the downstream DAE. This is exactly the function of coincidence detector in the cortex, to distinguish the coincident spikes from averaged dispersed spikes for downstream processing.
>
> In sum, we term eq 6 as (generalized) CD because it captures the core feature of CD (narrow window and low threshold) and also function as a coincidence filter to encourage the emergence of synchrony during learning (Section 4.2). Therefore, it captures more about the picture, either for model itself or bio-correlates of the model. Besides, the simple formulation of eq6 could naturally be extended to consider a local region of neurons (but it is not necessary in this model), so that the coincidence of multiple neurons is explicitly taken into account. For example, in redundant/coarse coding scheme, there could be a column of neurons of similiar response property at each location of SCS, instead of a single neuron at each location. In this case, each CD can be extended to be a non-linear low-threshold function of this column of neurons instead of a single $s_i$ so that multiple coincident spikes is taken into account (coincidence spikes indicate the confidence of the presence of objects/features).
>
> __A3. The response to the refractory period__.
>
> While the top-down modulation by DAE provides the positive feedback (reinforce the original pattern), which is needed to construct a synchrony state. But to group multiple objects, we need different synchronized neuronal groups to alternate. Without competition (negative feedback), the winner pattern will dominate forever. Here, refractory period enforces a hard temporal competition of neuron itself: if the neuron fires at some time, it cannot fire later. Therefore, different possible ‘timings’ compete with each other to fire the spike for each neuron. Since the objects are grouped at different timings, each timing point is a _possible_ candidate grouping ‘slot’. Refractoriness provides competition among these ‘candidate timing slots’, which we call temporal competition. Since refractory neuron is a structural constraint, it acts as structural bias to avoid WTA pattern and encourage synchronized neuronal groups to alternate. It is notable that, here, refractoriness is a special type of self-inhibition. In the brain, similar temporal competition/self-inhibition could possibly be provided by inhibitory neurons, not restricted to refractory mechanism, which is also plausible to induce various types of oscillations.
>
> __A4. The response to the concern on simulation period__.
>
> We provide the detailed explanation of simulation length in A.9.1 of the Appendix: ‘simulation length’ and ‘additional results’ (also see Figure5/6 and Table2 in the appendix).
>
> During training, the simulation length is 100 (Table2 in appendix). We keep the simulation length relatively small because it is more efficient to backpropagate the gradient and it saves the training time. During testing/visualization, the model indeed tends to converge during the first 100 time-steps (Figure 5 in the appendix), but even longer simulation time could allow the model to converge more completely. That is why we use 3-times longer length for testing and visualization. On the other hand, it also confirms the stability of the model.
>
> __A5. The response to the concern on figure5__.
>
> Each small-square is a snapshot of SCS population activity (marked as ‘s’) or the attention feedback of the same dimension (marked as ‘$\gamma$’). Each row of small squares is last-10 time-step simulation of the population activity (after the epoch). More detailed discussion can be found in A.9.3 of the appendix.

---

### Official Review · Reviewer_fSiP · 2023-07-07

**Soundness:** 3 good
**Presentation:** 3 good
**Contribution:** 3 good
**Rating:** 7
**Confidence:** 3

**Summary:**

The research topic of this study is development of a model that can learn to group (segment) the pixels in an image into objects in an unsupervised manner and in a way to enable systematic (combinatorial) generalization with respect to the number of objects. Toward this goal, the authors proposed a model, named GUST, by taking inspirations from the mechanism of human brain. The proposed model consists of two types of neural networks, spiking neural network and denoising autoencoder, that implement an iterative bottom-up/top-down processing together with several other brain-inspired components. The authors selected a simple loss function corresponding to image denoising for unsupervised training, and proposed a strategy for end-to-end gradient-based traing and also a clustering method for grouping (segmentation). The effectiveness of the proposed method was tested with synthetic images containing simple 2D objects. The proposed method has shown superior generalization performance over two competitors with respect to the difference in the number of objects during training and testing. The characteristics of the learning process and role of each component were also empirically studied.

**Strengths:**

Unsupervised segmentation (grouping) itself is a challenging computer vision task, and also can be the important first step in more complex visual tasks. Systematic (combinatorial) generalization to new situations is a hallmark of human intelligence yet to be achieved by machine learning models. The result that a model consists of spiking neural network and denoising autoencoder with several other brain-inspired components can be effectively trained to solve segmentation task in a systematic generalization setting in an end-to-end unsupervised manner will be of interest to the NeurIPS audience.

The ideas and implementations of the brain-inspired components are explained fairly well, and the actual code is accessible via the URL provided in the Appendix. The authors provide empirical analysis about the learning process and the effect of the brain-inspired components (ablation study) in addition to the simple performance report, which gives additional values to this study. Although the generalization performance is still not perfect and the experiments are conducted with simple synthetic images, this study shows an interesting research direction worth pursuing.
\# I took additional experiments provided in the Appendix into consideration when I evaluated the variety of  experiments.

**Weaknesses:**

A relatively weak point of this study is the difference from Zheng et al [38, in References for the main body of the paper]. Although the proposed model (GUST) is advanced compared to DASBE [38] in multiple aspects as stated in Section 5 (Related work), the core idea of combining spiking neural network and denoising autoencoder in brain-inspired manner and using the temporal coherence for grouping is proposed in DASBE.

[38] Hao Zheng, Hui Lin, Rong Zhao, and Luping Shi. Dance of SNN and ANN: Solving binding problem by combining spike timing and reconstructive attention. In S. Koyejo, S. Mohamed, A. Agarwal, D. Belgrave, K. Cho, and A. Oh, editors, Advances in Neural Information Processing Systems, volume 35, pages 31430–31443. Curran Associates, Inc., 2022.

**Questions:**

Question

Are there any other differences between your study and Zheng et al [38] than the six aspects stated in Section 5  (Related work)?

Major suggestions

1. Describing further differences from  Zheng et al [38] (if any)
If the answer to the above question is yes, it is beneficial to describe the additional differences in the paper (either in the main body or in the Appendix). The differences are not necessarily to be about GUST and DASBE but can be about other aspects of the studies (papers). If any of the already stated six aspects can be further detailed to emphasize the novelty of this work, it would also be beneficial.

1. Clarifying explainability issue
In the paragraph staring at line 73, four challenges are stated; the last one is explainability of the representation. However, it seems (to me) that the answer/discussions about this one is not clearly stated in the paper, at least compared to the other three. It would be better if  this foreshadowed issue could be highlighted with explicit keywords like "explainability"or "explanable" in the latter part of the main body of the paper.


1. Adding pointers to the Appendix in the main body of the paper
Although the Appendix of this paper contains rich contents, currently it is mentioned only three times in the main body (without section numbers).  Additional appropriate pointers to the Appendix in the main body would be beneficial for the readers.  (For example, I had a concern about the limited variety of inputs in the experiments when I first read the main body, but later I realized that results with other inputs are provided in the Appendix.)

Please also refer to the Weaknesses section above and Limitations section below.

Minor questions and suggestions

1. Figure 2 (a) lacks $x$ and Figure 2 (b) lacks $\tilde{x}$. It is better to connect these two in figures.
1. Is $\tau_w$ in line 177 a typo of $\tau_1$? If not, what is $\tau_w$?
1. The code availability is better to be stated in the main body of the paper.
1. Please review the descriptions in References. For example,  von der Malsburg 1994 [6 in References for the main body] and von der Malsburg 1994 [11  in References for the main body] lack information about what kind of publications they are,  and Engelcke et al [53 in References for the main body] was accepted at ICLR 2020.

**Limitations:**

The limitations are detailed in Appendix A.1, but the existence of this section is not indicated in the main body of the paper. It would be better if the authors mention the section at an appropriate place in the main body when they revise the paper.

In addition to the limitations already stated, I think the current study is also limited in the following two points.

1. The objects used in the experiments are all 2D. (Experiments with images resulted from 3D objects would be preferable to be used in the future work.)
1. The background is void. (If the background also consists of objects, the task becomes more difficult. This situation would be also preferable to be tackled in the future work.)

If these are correct, it would be nice to mention them as well.

---

> ### Author Rebuttal · Authors · 2023-08-06
>
> __A1. Major response__ to the concern about the difference with Zheng’s work.
>
> We will fist detail the six aspects in Section 5 and then point out more technical differences.
>
> First of all, the two papers are __answering different questions__ of different levels, not just advanced. Intuitively, the Zheng’s work focuses on how to __design__ a clustering program (like EM) while ours focuses on how to __learn__ the program from blank slate. Therefore, Zheng's focuses on dynamics only, but we take learning process into account. To some extent, from design to learning is an important step made by machine learning/deep learning, which is not obvious at all. It is especially the case here. While the architecture and final phenomenon is similar between the two works, which is actually desirable, the underlining mechanism is largely different, which is discussed in Section 4.2,4.3 in the main text (also Figure 11 in appendix). For example, _during training_, how blank slate DAE evolves so that: initial global random guess breaks the symmetry and tends to attend on a local spatial area, followed by further capturing the texture details. _The new insight_ we provide is that, the _symmetry-breaking behavior_ (grouping each single object into alternative synchronized assemblies, given multiple objects) can be learned with a succinct _symmetry-preserving_ loss function (reconstructing the whole multi-object scene). The Zheng’s work did not provide insight at this level. Additionally, evaluation of the __generalization__ is also not accounted for in Zheng’s work, since it does not have a complete learning scheme.
>
> Second, due to the difference of answered questions, the __interpretation of the biological factors__ (or network architecture) is different. In Zheng’s work, factors like delayed coupling and spiking dynamics function only as _shaping the network dynamics_ so that the convergent state is the synchrony. However, in our work, we provide additional new insights that the factors like narrow time window, refractoriness, etc, could even act as inductive bias to _bias the learning_ process (Figure2 in Appendix). The Zheng’s work does not provide insight about network architecture at this level.
>
> Third, due to different interpretation, the __underlining hypothesis about how brain could implement such a model__ is totally different. Zheng’s work must assume a _pre-wired_ cortical bottom-up/top-down pathway as a pre-trained single-object DAE, which is constructed during development, determined by gene or evolution. But how each potential single object be accounted for by pre-wiring? In contrast, our work relaxes this assumption and suggests that the brain could _gradually learns_ such a model in a way consistent with predictive coding during the life even after the development.
>
> Forth, to some extent, the DAE module (Section 3.3) in our work is not 'really' a DAE (it is not trained separately/explicitly to denoise), but a general encoder-decoder structure. The _denoising_ is achieved by the _whole system_ during iteration instead of the DAE module its own (Zheng's). We term it as DAE to provide a more intuitive picture of the architecture. However, __the nature of the architecture__ is different between Zheng’s and ours.
>
> We believe these aspects conceptually distinguish this work from Zheng’s, not just technically advanced. In addition to these conceptual-level novelty, there are also technical difference. Some are introduced in Section 3.4,3.5 in the main text, including the __novel loss function__ and dealing with the __gradient__ along delayed path and refractory period.
>
> Besides, one major difference on study is that we provide a __new quantitative evaluation scheme__ of the grouping representation (more efficient). In Zheng’s work, it uses K-means to cluster the spike trains (to compute the AMI), which is hard to explicitly distinguish time coding from rate coding (A.13.2 SI). In our work, we resolve this problem by explicitly taking timing code into account when clustering. That is why we introduce K-medoids and VP-metric in Section 3.6. SynScore also differs and is more rational. Therefore, the quantitative scores have different interpretations between the two works. This is the technical novelty of the evaluation.
>
> In sum, while the basic idea of combining spiking neural network and denoising autoencoder is shared between Zheng’s and ours. They differ in (1) motivation (representation/dynamics or learning/generalization) (2) nature/interpretation of the architecture (3) underlining mechanism/bio-picture (4) evaluation scheme. These stress the nolvety of this paper.
>
> __A2. For explainability__, actually, to analysis the grouping, the representation must be explainable. As a result, by clustering/visualizing the spiking pattern in Section 4, we make the representation in SCS explainable.
>
> __A3. For advices on revision__, including explainability/pointers/figures/code/reference/limitation, we will take these helpful advices into account when we revise the paper.
>
> __A4.__  $\tau_w$ is the internal model parameter while $\tau_1$ describe the externally observed synchrony (Figure2 in main text). They are relevant but not limited to be identical
>
> __A5. For future work on 3D and background__. Actually, this is one of our ongoing works (group in clever dataset, which is 3D objects with varied background). We find that the two limitation is firstly caused by the binary SCS in the pixel space because 3D objects should be grey-scale at least and general background is also difficult to be accounted for in binary image. As a result, we decide to implement SCS as a __binary hidden layer__ so that there are no constraints of the input type (binary/gray-scale/colored). The preliminary results of the ongoing work show that such change of network architecture preserve the basic mechanism and grouping/synchrony in hidden layer, still emerge. We will further study this situation as future work

---

> > ### Comment · Reviewer_fSiP · 2023-08-18
> >
> > Thank you very much for the detailed answers. My questions and suggestions have been adequately responded. Assuming that the contents of the rebuttal will be appropriately reflected in the revised paper, I raised the score for Presentation from 2 to 3, that for Contribution from 2 to 3, and that for Rating from 6 to 7 (about the last one, considering the point that this work is related to machine learning, computer vision,  and neuroscience).
> >
> > \# About **A4**, it is further beneficial if the equation (6) is visualized in a figure with the clarification of the meaning of the time window ($\tau_w$).

---

> > > ### Author Response · Authors · 2023-08-18
> > > **Response to the Reviewer fSiP**
> > >
> > > We thank the reviewer for recognizing the contribution of the work and providing the helpful suggestions.
> > >
> > > We are aware that $\tau_w$ should be further explained in the main text.
> > >
> > > As a result, we slightly modified the eq (6) as:
> > >
> > > $s'_i(t) = CD(s_i(t),...,s_i(t-\tau_w+1))$
> > >
> > > $ =1, \sum_{t'=0}^{\tau_{w-1}}{s_i(t-t')} >= 1$
> > >
> > > $= 0, \sum_{t'=0}^{\tau_{w-1}}{s_i(t-t')} < 1$
> > >
> > > where $\tau_w=2$ in the current model$.
> > >
> > > In this way, $\tau_w$ is explicitly included in the eq(6). and we clarify the meaning of $\tau_w$.
> > >
> > > We will also try to add $\tau_w$ in Fig2(b) in main text  without inviting too much details.

---

> > > > ### Comment · Reviewer_fSiP · 2023-08-21
> > > >
> > > > Thank you very much for the additional consideration. Now the meaning of "$\tau_w = 2$" has become clearer by the new equation (6). About the visualization, it can be done in Appendix if there is not enough space in the main text. Anyway, thanks again for your earnest attitude for revision.

---

### Official Review · Reviewer_dadF · 2023-07-09

**Soundness:** 3 good
**Presentation:** 3 good
**Contribution:** 2 fair
**Rating:** 5
**Confidence:** 4

**Summary:**

This paper tackles the challenge of grouping information from individual visual elements into whole perceptual units, i.e., how compositional generalization is achieved, through proposing a GUST (Grouping Unsupervisely by Spike Timing network) that leverages spiking synchrony for grouping. This framework introduced a spike coding space (SCS) and a denoising autoencoder (DAE) to control for temporal grouping, enabled representing scenes of combinatorially different structures in simulated datasets.

**Strengths:**

1. This paper combined insights from the neuroscience literature including correlation brain theory and temporal coding to enable grouping mechanisms in unsupervised learning.
2. The paper tackles a non-trivial challenge wrt designing a spike timing neural network properly, given that spiking is non-differential, requirement of high temporal precision, and interpretations of representations. Specifically, the proposed algorithm enables learning directly from multi-object inputs. It also enables a gradient-based training framework. The paper further leveraged non-linear metrics to evaluate grouping performance based on neuronal synchrony. Overall, the paper proposed elegant solutions for these challenges.
3. The paper examined the effect of biological constraints in their algorithm by providing an ablation experiment, and demonstrated  superior grouping performance in novel scenes than baseline models RC-bind and Slot Attention.

**Weaknesses:**

1. I'd love to see stronger evaluation benchmarking with more metrics, given that unsupervised clustering often gives different results based on how they are tuned and often is biased by noise. The authors leveraged the adjusted mutual information (AMI) to evaluate the grouping quality, and used Silhouette score (SC) to evaluate inner-cluster coherence level. For example, SC is generally higher for convex clusters, and it is known that different metrics sometimes choose different preferred method/algorithm in noisy clustering cases.   Despite AMI adjusts for chance, AMI is high when there are pure clusters in the clustering solution (see [ref](https://jmlr.csail.mit.edu/papers/volume17/15-627/15-627)). It is unclear of the dataset clustering distribution to evaluate potential bias in their metrics. Specifically, I'd like to see more analysis on how their benchmarking performance varies across different random seeds, number of clusters, etc. Moreover, I'd recommend to add benchmarking results using ARI (adjusted rand index), to provide additional information.
2. The grouping evaluation focused on clustering different objects. This is very limited to persuade the readers on how their algorithm demonstrated superior compositional grouping. I'd recommend to add more evaluations on different aspects of compositions. For example, instead of classify different shape objects, is the proposed algorithm able to learn compositional structures in an object. For example, can different body parts of a complicated object being learned and segmented?
3. A major challenge of unsupervised learning on multi-object inputs is when objects are overlapped with different degrees, or collapsed with different noise levels. There is no analysis to quantify the distribution in datasets. Although image in Figure 4 indicates a partial overlap of triangle and square shapes, Figure 5 suggests those shapes are mostly clearly separated in space. It is thus unclear if the proposed algorithm can deal with grouping and unsupervised learning in these scenarios.
4. The whole idea of leveraging temporal coherence as a way for learning generalization and classification is not new, and has been explored in the literature. For example, this [paper](https://journals.plos.org/ploscompbiol/article?id=10.1371/journal.pcbi.1005137) shows that a spiking network driven by input timing dependent plasticity (ITDP) could perform visual classification task well and to generalize to unseen datasets. This [paper](https://arxiv.org/pdf/1807.10936.pdf) shows a hierarchical spiking architecture in which motion can selectively emerges in an unsupervised fashion. Additionally, how combinatorial coding can emerge in the cortex also has been discussed in [paper 1](https://www.ncbi.nlm.nih.gov/pmc/articles/PMC2693376/), [paper 2](https://www.princeton.edu/~wbialek/our_papers/osborne+al_08.pdf), and [paper 3](https://www.ncbi.nlm.nih.gov/pmc/articles/PMC4605134). This paper did not cite these relevant work, nor discussed how their proposed algorithm outperforms or links with these previously proposed combinatorial coding mechanisms. It is thus difficult to evaluate the novelty and whether this paper brings additional new insights in bio-inspired algorithms (see more comments in Limitations).

**Questions:**

See corresponding questions and suggestions in the Weaknesses section.

**Limitations:**

This paper proposed a brain-inspired algorithm and assumed representations of combinatorially different structures happen within the same spiking neural network for classification. However, we already know this is not entirely true in the mammalian brain circuits. Often there is a clear hierarchical and modular structure, where low-level features were represented in early-stage cortical regions, which further enables compositional encoding. From this aspect, the paper seems limited in bringing more insights for biological interpretations. From the aspect of computational algorithms, the paper does not demonstrate superior performance compared to other SOTA unsupervised learning through deep neural networks. Therefore, this paper seems limited in terms of bringing higher impacts for both neuroscience and machine learning fields. I would suggest the authors to elaborate the limitations from biological interpretations, and the future directions of this work.

---

> ### Author Rebuttal · Authors · 2023-08-05
>
> __A1 Benchmarking analysis and selective bias__: See __A1, A2 in global response__.
>
> __A2. Concerns on hierarchical case (brain organization / hierarchical grouping)__.
>
> First,  we consider combinatorial generalization of _single-level grouping_ in this paper, where varying object number seems to be complete for composition, instead of _compositional grouping_.
>
> Second, the GUST architecture in this paper serves as the basic building block of more complex systems for more complex binding (bigger picture). Therefore, the GUST architecture is consistent with hierarchical and modular structure of mammalian brain and can be generalized to account for hierarchical grouping of ‘compositional structures’: GUST is each ‘cortical column’ to group features that belong to body / part of respective levels.
>
> Third, we are actually pursuing this idea as an ongoing work: hierarchical organization of multiple GUSTs for hierarchical grouping like the brain ('early stage for low-level features' as reviewer suggested), which we term as _representing the part-whole hierarchy of the visual scene_. Specifically, each GUST module serves as 'column' of each level, and different levels of GUSTs (characterized by their own time scale constants, fast for low-level GUST and slow for high-level GUST) are organized in a hierarchy (__Fig 6 a,b in global response__). The emerged synchrony (to represent part-whole) in turn is also of different time scales, fast one for part and slow one for whole, nested with each other like gamma-theta coupling. Preliminary results are shown in __Fig.6 c in global response__. However, as argued in A.1 (line108) in SI, this paper prefers to keep architecture minimal and general and leave ‘composition grouping’ (reviewer suggested) as future works. For other aspects of grouping / generalization results, see A.12 in SI
>
> __A3 Concerns on overlap__.
>
> The grouping is robust w.r.t. the overlap or noise since the attractor dynamics provides a completion process. For example, top-down feedback is a completion of the bottom-up noisy firing (Figure 11 in SI). See __A3 ~ A6 in global response for more__.
>
> __A4 Concerns on  contribution to machine learning / neuroscience literature__.
>
> First of all, unsupervised grouping is a hard problem in deep learning / CV field (as reviewer fSip correctly points out, also stressed in ref[7] in main text) and it is also a fundamental operation for human brain, still lacking a clear mechanism. Therefore, combining the two sides on the one hand provides a general solution of basic computational problems in machine learning, and on the other hand, provides a systematic algorithmic level understanding of various biological structures/phenomenon/factors (like top-down feedback in cortex, delay-coupling, refractory, synchronized assembly, gamma oscillation, etc), bridging the levels (See A.5 in SI). As far as we know, both contributions bear the novelty.
>
> Secondly, for related work on bio-algorithms, the reviewer seems missed the core problem in this paper, grouping, which motivated the whole story. We do not exploit coherence for general learning / classification / generalization (as the IDTP paper and optical flow paper shows), but to solve the grouping problem in neural network, which is a problem at more basic level: representation (learning and generalization is relatively secondary). Therefore, they are totally different algorithms. Besides, aforementioned two papers use feedforward architecture without feedback and therefore coherence is not an emergent property through recurrent dynamics. Also, the dynamical nature of their ‘coherence’ is different from ours (though apply the same term). Our model has switching synchronized groups, which is non-trivial (need symmetry-breaking, __new insight__) and essential for grouping (__new insight__), while the two related papers only consider a single synchronized group for classification. Therefore, they differ from ours on at least (1) why coherence (2) what coherence (3) how coherence.
>
> Thirdly, for combinatorial code, actually it is an elegant abstraction of spike code beyond rate code. Here, we specifically focus on synchrony code itself. The presence of synchrony code in cortex and their contribution to the grouping function has been cited as [17~24,26-28] in main text. There is such a wide range of supports in neuroscience that we cited the most relevant ones. Besides, the 2 works by William Bialek are appealing theoretical assessment of information contained in the combinatorial code, with a _descriptive_ model to formulate the probability,  instead of providing a _computational mechanism_ of how combinatorial code can emerge (as the reviewer improperly stated) and how they function to solve problems. Our work focuses on the latter two sides (__new insight__). Even so, we share a common spirit on additional information beyond rate code and they may potentially inspire our future works like binding by polychronization.
>
> Fourthly, the CTC theory (old or new) takes the existence of coherence for granted and directly focuses on its contribution to communication, instead of the other way around:  communication also contributes to coherence. Therefore, it is not a closed framework. In our work, we provide a closed framework of (1) how coherence contributes to communication between DAE/SCS and (2) how communication between DAE/SCS also contributes to the emergence of coherence (Section 4.2 in main text, A.4.1,A.4.2 in SI). It is an iterative process instead of a one-way process in CTC (__new insight__).
>
> Lastly, we discussed limitation/future direction in A.1 of SI. As reviewer suggested, we will add more on biological limitations when we revise the paper, including the correlation-based plasticity like IDTP, more general combinatorial coding like polychronization, and hierarchical grouping in a hierarchical architecture. For concern of SOTA, 'slot attention' is one such recent SOTA on color image (See A.9.6 in SI).

---

### Author Rebuttal · Authors · 2023-08-08

We thank all the reviewers for spending their valuable time to carefully read the paper and writing reviews. The comments and suggestions are very valuable and insightful, which helps to revise the paper and motivate future directions. We take all suggestions carefully when we revise the paper.

Specific responses to each reviewer are provided in separate rebuttal panel and in this global response, we address several concerns that requires new figures or related to concerns of multiple reviewers.

___Response to questions on evaluation/benchmarking___

__A.1 (from reviewer dadF) Rationality of AMI and SC for evaluation, based on data distribution__.

On the one hand, _AMI / ARI are common-practice evaluation metrics to evaluate grouping_ (ref [14, 38, 48, 50, 51, 52, 53] in main text, discussed in A.13.3 in SI). On the other hand, we include background during both K- Medoids clustering and AMI evaluation (K=object number +1) since removing background introduces explicit prior of foreground-background segmentation (eg. what is object), which should also be considered for general grouping ability. Actually, during experiment, we found that confusing near-boundary region of Shapes with background is one reason that challenges the evaluation scores. In this case, the cluster size is relatively unbalanced because background tends to be the bigger cluster. Since AMI is more suitable for unbalanced situation, AMI is a more suitable metric than ARI, which fits more for balanced case (eg. if background is explicitly removed for evaluation). Also see __Fig 2__ right in pdf.

For SC, we visualize the low-dimensional structure of clusters through PCA in the metric space of spike trains induced by VP-metric (__Fig 5__ in pdf). It is shown that in both non-overlap and overlap cases, clusters are well-centered and seperated ('convex' in metric space), so that SC is suitable to evaluate the coherence level.

__A.2 (from reviewer dadF) More benchmarking results w.r.t random seed, number of clusters, AMI vs ARI__.

First, we have used 5 random seeds to evaluate the averaged AMI score in figure 6 in main text and the error bar is neglectable (Table3 in Appendix). Thus, the averaged AMI within large dataset is not influenced by uncertainty of AMI on each single sample (shown in __Fig 2__ in pdf), and there is no bias.

Second, the benchmarking in figure 6 in main text (of different object number) already indicate different number of clusters (K=object number+1). As far as we understand, it is exactly what reviewer asked for. __Fig 4__ in pdf is a more detailed benchmarking with varied object / cluster number.

Third, as explained in A1, AMI is a more suitable metric than ARI in our case. We also provide benchmarking on ARI (__Fig 4__ in pdf) for supplement.

Lastly, we provides additional benchmarking results w.r.t different overlapping degree in __Fig 4__ in pdf (both AMI and ARI).


__A.3 (from reviewer 9PtM). Compare the sensitivity among models (Performance vs Overlap ratio)__.

Please see __Fig 4__ in pdf (upper), where comparison is made for '3 train objects and 3 test objects' case (major case). Due to page limit, it is implausible to provide all 9 combinations (2/3/4 train objects X 2/3/4 test object) as Figure 6 in main text.

___Response to questions on Overlap / Noise___

__A.4 (from reviewer dadF) Analysis of the distribution of overlap in dataset__.

Figure 5 in main text do not suggest shapes are mostly clearly separated (the reviewer argued), because this image is just one randomly selected example. Besides, the example in Figure 4 in main text, in contrast, has partial overlap, which is much more common (__Fig1__ in pdf). We visualize the distribution of overlap in __Fig 1__ in pdf.

__A.5 (from reviewer dadF) Whether GUST can deal with overlaps__.

Figure 4 is one example, which suggests that GUST can deal with partial overlap through its denoising mechanism. Actually, in Shapes dataset, overlap is quite common (__Fig 1__ in pdf). We provide systematic quantitative analysis of overlap effect in __Fig 4__ in pdf.

__A.6 (from reviewer dadF) Whether GUST can deal with noise__.

The DAE in GUST can denoise the stochastic firing pattern in SCS through its feedback. However, input noise will linearly affect the spike firing, since in current implementation, input _hard gates_ the spiking neurons. If grouping is decoded from SCS ($s(t)$), the grouping will be linearly affected, but if grouping is decoded from feedback ($\gamma(t)$), it is robust to noise. It is plausible that adding horizontal spatial connection in SCS (future work) will provide pattern completion in SCS based on proximity law (Gestalt principle), when given noisy input.

___Preliminary results on hierarchical grouping___

__A7. (reviewer dadF) How GUST can be extended to hierarchical grouping__.

Schematic architecture and preliminary result of hierarchical grouping (part-whole hierarchy) is shown in __Fig 6__ in pdf.

__A8. (reviewer 9PtM) How uncertainty encoded in neuronal coherence can be used  (for hierarchical grouping)__.

Here we provide an example of one ongoing work (part-whole hierarchy).

The overall architecture is shown __Fig6__ a,b in pdf. To represent the part-whole hierarchy of a visual scene, it is efficient to seach for solutions in a divide-and-conquer way: whole first and then part (possible brain strategy). Beginning with high temporature, high-level grouping achieves the high-level coherence (slow oscillation, coarser spatiotemporal structure) while low-level still remains unstructured/ random; __this high-level coherence (slow oscillation) indicates high certainty in whole-level grouping and is readout by inhibitory neurons to lower the temporature (decrease the excitability) so that low-level grouping is further achieved__ (faster oscillation, finer spatiotemporal structure), _conditioned by high-level grouping_. Overal, coherence acts as __internal indicator__ of a good solution with high confidence.

---

### Decision · Program_Chairs · 2023-09-21

**Decision:**

Accept (poster)

**Comment:**

This paper addresses the complex task of aggregating information from individual visual elements into coherent perceptual units, a phenomenon known as compositional generalization. The proposed solution, GUST (Grouping Unsupervised by Spike Timing network), harnesses spiking synchrony to achieve grouping. The framework introduces a spike coding space (SCS) and employs a denoising autoencoder (DAE) to regulate temporal grouping, thus enabling the representation of scenes with diverse structures in simulated datasets.

Drawing from neuroscience literature, the paper integrates concepts such as correlation brain theory and temporal coding to establish grouping mechanisms in unsupervised learning. The challenge of effectively designing a spike timing neural network is tackled, considering the non-differential nature of spiking, the need for precise temporal encoding, and interpretability of representations. The study examines the algorithm's resilience to biological constraints through an ablation experiment, showcasing superior grouping performance in novel scenes compared to baseline models RC-bind and Slot Attention.

We think there is a good novelty and contribution to the community.